# THINK-BENCH: EVALUATING THINKING EFFICIENCY AND CHAIN-OF-THOUGHT QUALITY OF LARGE REASONING MODELS

## ABSTRACT

Large reasoning models (LRMs) have achieved impressive performance in complex tasks, often outperforming conventional large language models (LLMs). However, the prevalent issue of overthinking severely limits their computational efficiency. Overthinking occurs when models generate excessive and redundant tokens that contribute little to accurate outcomes, especially in simple tasks, resulting in a significant waste of computational resources. To systematically investigate this issue, we introduce Think-Bench, a benchmark designed to evaluate the reasoning efficiency of LRMs. We also propose novel efficiency metrics and conduct a comprehensive evaluation of various LRMs across multiple dimensions, including the reasoning process, outcome quality, and chain-of-thought (CoT) characteristics. Our analysis reveals that most LRMs exhibit overthinking in handling easy questions, generating unnecessarily lengthy reasoning chains. While many LRMs demonstrate high CoT quality, several suffer from low efficiency. We hope that Think-Bench[1] can serve as a robust foundation for advancing research into LRMs.

## 1 INTRODUCTION

Recent advances in LLMs have led to remarkable progress in text generation and question answering (Grattafiori et al., 2024; Guo et al., 2025; Yang et al., 2024). However, in structured, multi-step reasoning tasks, LRMs still face persistent challenges in efficiency and reliability (Wang et al., 2024; Chen et al., 2025). With the growing use of CoT prompting and test-time scaling strategies, models often produce excessively long or repetitive intermediate steps, a phenomenon we refer to as *overthinking* (Chen et al., 2024; Guo et al., 2025). On simple problems, overthinking introduces substantial computational overhead while providing little to no improvement in accuracy. This inefficiency increases deployment costs such as latency and resource consumption, and it also undermines interpretability, since verbose reasoning may obscure the logical validity of each step.

Existing multi-disciplinary benchmarks such as MMLU (Hendrycks et al., 2020a) and GPQA (Rein et al., 2024) primarily evaluate

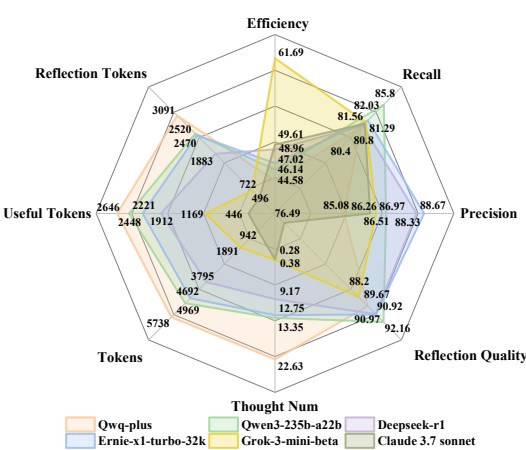

Figure 1: **The performance of various LRMs on Think-Bench.** The results suggest that these prominent LRMs face a challenge of overthinking.

---

[1]The anonymous links to the code and dataset: `https://anonymous.4open.science/r/Think-Bench-anony-6866`

Figure 2: **Overview of Think-Bench.** Our benchmark contains a comprehensive efficiency evaluation framework with curated datasets across three categories.

reasoning models by their final-answer accuracy, while the reasoning process itself remains largely unexamined. This outcome-centric evaluation paradigm fails to capture redundant steps, logical inconsistencies, and repeated verification, thereby limiting our understanding of LRMs' actual reasoning abilities and computational costs (Jiang et al., 2025; Zheng et al., 2024). Although several recent studies have explored CoT quality or robustness (Jiang et al., 2025; Zhou et al., 2024), a comprehensive framework that evaluates both reasoning efficiency in terms of token and time cost and step-level correctness of reasoning chains is still missing (Wang et al., 2025).

To address this gap, we introduce **Think-Bench**, a benchmark for evaluating both thinking efficiency and CoT quality. Think-Bench covers mathematics, physics, and chemistry with 1,375 problems, each annotated with human-curated key reasoning steps. These annotations enable fine-grained alignment and rigorous evaluation of model-generated reasoning. Based on this dataset, we design a set of complementary efficiency metrics, including total tokens, first-correct tokens, efficiency, reflection tokens, and number of thoughts, as well as two interpretable CoT quality metrics: recall and precision. Together, these metrics provide a comprehensive framework for analyzing reasoning behavior in terms of speed, cost, and correctness (Chang et al., 2024; Xia et al., 2024).

We conduct extensive experiments on 11 representative LRMs, covering both proprietary and open-source systems of varying sizes and design objectives. The results reveal several important findings. First, most models display pronounced overthinking on simple problems, producing large amounts of redundant reasoning for only marginal or no gains in accuracy. Second, larger models tend to achieve higher CoT quality but at the expense of reduced efficiency. Third, certain models, such as Grok-3-mini-beta, demonstrate a more favorable balance between efficiency and quality. Based on these findings, we highlight several promising directions for improving reasoning efficiency, including dynamic reasoning-path design, early-exit mechanisms, and path selection strategies that reduce reasoning costs without sacrificing reliability.

Our contributions are threefold:

1. We present **Think-Bench**, a multi-disciplinary benchmark with fine-grained step annotations for evaluating reasoning efficiency and CoT quality.

2. We design a systematic evaluation policy and introduce novel metrics that enable quantitative analysis of overthinking and reasoning reliability.

3. Through comprehensive experiments on 11 LRMs, we provide the first large-scale empirical study of efficiency-quality trade-offs and suggest promising directions for future research and optimization.

## 2 DATASET CURATION

### 2.1 DATA OVERVIEW

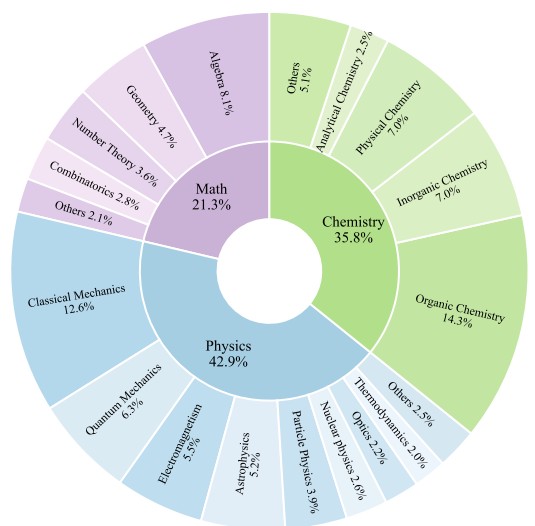

Figure 3: Category and Subcategory Distribution of Think-Bench.

Table 1: Key Statistics of Think-Bench.

| Statistic | Number |
|---|---|
| Total questions | 1375 |
| -Choice questions | 929 |
| -Free-form questions | 446 |
| -Math questions | 293 |
| -Physics questions | 590 |
| -Chemistry questions | 492 |
| Total key step annotation | 13311 |
| -Average inference step | 9.68 |
| Maximum question length | 1893 |
| Maximum answer length | 372 |
| Average question length | 422.42 |
| Average answer length | 7.59 |

As shown in Figure 2, Think-Bench is a dataset specifically designed to evaluate the thinking efficiency and the quality of CoTs of LRMs in complex reasoning tasks. This dataset comprises 1,375 carefully selected and organized data samples, covering three core subjects: mathematics, physics, and chemistry. Within each subject, the number of simple questions is approximately equal to the number of difficult questions. The data sources are diverse, drawing from multiple academic datasets, including MMLU (Hendrycks et al., 2020a), Math500 (Hendrycks et al., 2020b), AGIEval (Zhong et al., 2023), AIME (Veeraboina, 2023), GPQA (Rein et al., 2024), SciKnowEval (Feng et al., 2024), and UGPhysics (Xu et al., 2025b).

### 2.2 DATA COLLECTION

During the construction of Think-Bench, we aggregated questions from multiple authoritative, publicly available datasets. The distribution of the Think-Bench across different disciplines is shown in Table 1. To ensure fairness, all samples were selected randomly. After the selection process, we conducted a systematic data cleaning and verification procedure to remove duplicate and invalid entries. The final dataset consists of 1,375 data points, after which we carried out the data annotation work. This benchmark covers the core disciplines of mathematics, physics, and chemistry, which inherently require structured and multi-step reasoning. Therefore, it provides a robust and rigorous foundation for evaluating the performance of reasoning models. Detailed statistics regarding the data composition can be found in Figure 3 and Table 1.

### 2.3 DATA ANNOTATION AND REVIEW

To systematically evaluate the CoT reasoning capabilities of LRMs on reasoning tasks, we implement a fine-grained annotation framework for key reasoning steps across all questions. In this framework, key steps are defined as the minimal set of indispensable logical components that are required for deriving the correct answer. Redundancies that do not involve essential mathematical or conceptual operations are explicitly excluded from the scope of key steps.

The annotation process adheres to a strict two-stage pipeline, with each stage designed to ensure the reliability and accuracy of the annotation results:

- Preliminary reference generation: For each question, Claude 3.7 Sonnet was employed to generate an initial CoT reasoning chain (see Prompt 1 in Appendix I). It is important to note that this initial chain served solely as a guiding reference for subsequent annotation procedures.

- Manual review and annotation: A team of graduate-level researchers with expertise in mathematics, physics, and chemistry manually reviewed, revised, and extracted reasoning steps. To ensure rigor, questions were assigned according to each annotator's domain expertise. In addition to identifying key steps, annotators also verified the assigned difficulty levels and corrected them when necessary, and further classified each question into a subdiscipline to support fine-grained domain-specific evaluation of LRMs.

For questions with multiple valid solution strategies, all consistent reasoning paths were included to capture the diversity of real-world problem-solving and enhance the generalizability of LRM CoT evaluation.

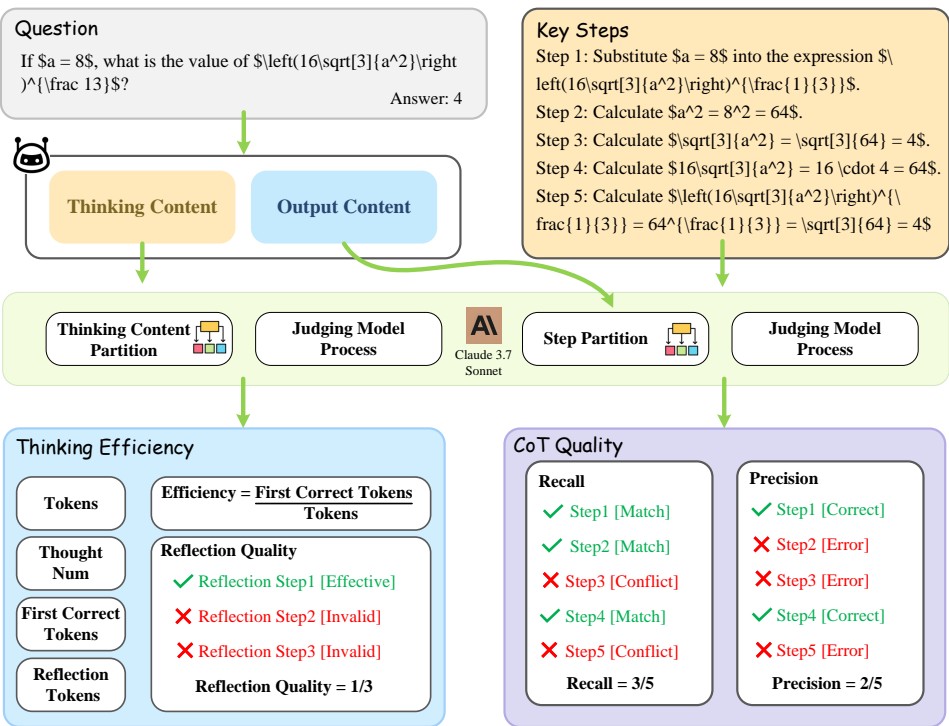

Figure 4: Illustration of Thinking Efficiency and CoT Quality Evaluation.

## 3 EVALUATION STRATEGY

Recent studies have revealed that LRMs frequently exhibit low reasoning efficiency (Sui et al., 2025; Chen et al., 2024). However, there is currently a lack of systematic benchmarks to evaluate this issue comprehensively. A detailed analysis of LRMs' reasoning processes is crucial for understanding their efficiency limitations and underlying challenges. Furthermore, existing benchmarks primarily assess the final answers to reasoning questions, neglecting the intermediate CoT steps. To bridge this gap, we propose a novel benchmark that jointly evaluates both the efficiency and quality of reasoning CoTs, thereby enabling a more holistic assessment of LRMs' reasoning capabilities.

### 3.1 THINKING EFFICIENCY EVALUATION

With the rapid development of LRMs, their ability to handle complex multi-step reasoning has significantly advanced (Xu et al., 2025a). Notable models like OpenAI-o1 (Zhong et al., 2024),

DeepSeek-R1 (Guo et al., 2025), and Qwen3 (Yang et al., 2025a) have attracted growing interest for their human-like capacity for extended, reflective reasoning. Through advanced long CoT and test-time scaling methods, these models iteratively evaluate multiple reasoning paths before finalizing answers (Chen et al., 2025; Muennighoff et al., 2025). However, as test-time scaling consumes increasing computational resources, a critical challenge has emerged in LRMs' inference behaviors: **Overthinking**. This refers to the model's persistent tendency to engage in excessive and repetitive reasoning, often producing reasoning chains that span hundreds of tokens even for simple tasks. While such elaborate verification is justified for complex problems, test-time scaling amplifies this behavior, causing unnecessary computational overhead and inefficiency during inference for simple inputs.

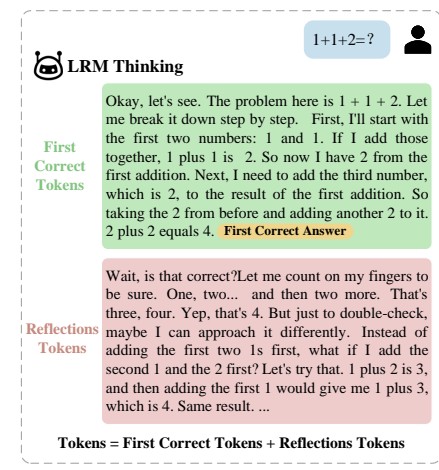

Figure 5: Example of Thinking Process Analysis in a LRM.

As illustrated in Figures 4 and 5, we propose six complementary metrics to systematically assess reasoning efficiency across token usage, inference dynamics, and reflective quality.

**Tokens** measure the total token count processed before final prediction, representing reasoning chain length and providing a fundamental basis for computational cost estimation.

**First Correct Tokens** measures the token count from reasoning initiation until the first occurrence of a correct answer. This metric evaluates the model's speed in reaching a valid solution during reasoning, where fewer tokens indicate faster correct convergence. The identification prompt is detailed in Prompt 5.

**Efficiency** is a normalized metric that refers to the ratio of first correct tokens to the total number of reasoning tokens. Formally, it is defined as:

$$\text{Efficiency} = \frac{1}{N} \sum_{i=1}^{N} \frac{\hat{T}_i}{T_i}. \tag{1}$$

where $\hat{T}_i$ denotes the number of tokens generated by the model before the first occurrence of the correct answer in its response, and let $T_i$ represent the total number of reasoning tokens for the $i$-th instance. If the model fails to produce a correct answer, we set $\hat{T}_i = 0$. A higher value of this metric indicates more efficient reasoning behavior. Concrete examples illustrating this calculation can be found in Figure 8 in Appendix D.1.

**Reflection Quality** measures the efficacy of the model's self-reflective reasoning, particularly after producing a correct answer. Not all reflective steps contribute meaningfully: some merely reiterate prior conclusions, while others may introduce erroneous revisions. We define a valid reflection as one that either (i) accurately identifies a prior error or (ii) provides new insights that confirm an earlier conclusion. Let $R$ represent the total set of reflective steps, and $R_{\text{valid}}$ denote the subset of valid reflections. The metric is defined as follows:

$$\text{Reflection Quality} = \frac{|R_{\text{valid}}|}{|R|}. \tag{2}$$

This score quantifies the efficacy of the model's reflection process, where higher values indicate more meaningful self-verification behavior as opposed to producing redundant or counterproductive content. The prompt used to guide this reflection process is provided in Prompt 6.

**Reflection Tokens** quantify the token count generated from the first correct answer to the conclusion of the reasoning process. This segment typically encompasses verification steps, reflective analysis, and conclusion restatements. Although such content may provide valuable insights, excessive length often signals reasoning inefficiency or unnecessary repetition.

**Thought Num** measures how often the model changes reasoning paths. This metric is estimated by counting discourse markers like "alternatively," "on second thought," and "wait a moment." A higher count may indicate instability in reasoning or a tendency toward exploratory behavior.

## 3.2 CoT QUALITY EVALUATION

As LLMs increasingly adopt CoT reasoning strategies, assessing the quality of their internal reasoning processes has emerged as a critical research challenge (Jiang et al., 2025). Existing evaluation approaches predominantly focus on final answer accuracy (Wang et al., 2019; Hendrycks et al., 2020b; Suzgun et al., 2022), while largely overlooking the validity and robustness of intermediate reasoning steps. To bridge this gap, we adopt a reference-based evaluation framework, inspired by MME-CoT (Jiang et al., 2025). Our proposed framework measures the reasoning quality from two interpretable dimensions: **Recall** and **Precision**.

Table 2: Evaluation of Nine Metrics on CoT and Efficiency in Think-Bench. Best performance in **bold**.

| Model name | Efficiency | Recall | Precision | Accuracy | Reflection Quality | Thought Num | Tokens | Useful Tokens | Reflection Tokens |
|---|---|---|---|---|---|---|---|---|---|
| Claude-3.7-sonnet | 49.61% | 81.29% | 86.26% | 93.89% | 76.49% | **0.28** | **942.82** | **446.09** | **496.73** |
| Deepseek-r1-distill-qwen-1.5b | 37.14% | 47.10% | 59.61% | 60.58% | 61.88% | 8.00 | 3734.49 | 1268.36 | 2466.13 |
| Deepseek-r1-distill-qwen-7b | 49.53% | 63.65% | 77.29% | 81.53% | 77.70% | 9.42 | 3504.76 | 1641.91 | 1862.85 |
| Deepseek-r1-distill-qwen-14b | 50.70% | 61.04% | 79.97% | 80.65% | 82.40% | 7.04 | 2814.75 | 1413.09 | 1401.66 |
| Deepseek-r1-distill-qwen-32b | 52.62% | 64.17% | 83.76% | 83.71% | 84.46% | 6.27 | 2697.70 | 1352.93 | 1344.77 |
| Deepseek-r1 | 48.96% | 80.80% | 88.33% | 93.82% | 90.92% | 9.17 | 3795.19 | 1912.12 | 1883.07 |
| Ernie-x1-turbo-32k | 47.02% | 82.03% | **88.67%** | 92.36% | 90.97% | 12.75 | 4692.21 | 2221.32 | 2470.89 |
| Grok-3-mini-beta | **61.69%** | 81.56% | 86.51% | 93.96% | 88.20% | 0.38 | 1891.34 | 1169.05 | 722.29 |
| Qwen3-235b-a22b | 46.14% | **85.80%** | 86.97% | **94.98%** | **92.16%** | 13.35 | 4969.05 | 2448.29 | 2520.76 |
| Qwq-plus | 44.58% | 80.40% | 85.08% | 90.76% | 89.67% | 22.63 | 5738.37 | 2646.73 | 3091.64 |
| Glm-z1-air | 47.41% | 80.16% | 83.18% | 91.92% | 89.17% | 9.80 | 3678.68 | 1775.07 | 1903.61 |

As illustrated in Figure 4, each CoT response is decomposed into multiple reasoning steps through the prompt detailed in Prompt 3.

$$R = \{r_1, r_2, \ldots, r_M\}.$$

To evaluate its quality, $R$ is compared against a pre-annotated reference set containing key reasoning components.

$$S = \{s_1, s_2, \ldots, s_N\}.$$

Each $r_j$ is judged for semantic alignment with any $s_i$, using Claude 3.7 Sonnet as a judge guided by consistent prompting instructions. The prompt designed to extract matching steps for computing recall and precision is provided in Prompt 2 and Prompt 3 of Appendix I. We define:

- $R_{\text{match}} \subseteq R$: the subset of reasoning steps in $R$ that correctly match at least one reference step in $S$.

- $S_{\text{covered}} \subseteq S$: the subset of reference steps that are successfully matched by at least one step in $R$.

Using $R_{\text{match}}$ and $S_{\text{covered}}$, we compute the **Recall** and **Precision** metrics as follows:

$$\textbf{Recall} = \frac{|S_{\text{covered}}|}{|S|} \quad (3)$$

$$\textbf{Precision} = \frac{|R_{\text{match}}|}{|R|} \quad (4)$$

Recall measures the extent to which essential reasoning steps are accurately captured in the LRM's output, reflecting the informativeness and comprehensiveness of the generated reasoning chain. In contrast, precision evaluates the correctness and relevance of the reasoning steps, penalizing any instance of inaccuracy or logical inconsistency.

## 4 EXPERIMENT

### 4.1 EXPERIMENTAL SETUP

**Evaluation Models** To systematically evaluate both the efficiency and quality of reasoning with CoT in LRMs, we select eleven representative models spanning diverse architectures and parameter scales. Our evaluation encompasses both proprietary and open-source LRMs. Specifically, we include Claude 3.7 Sonnet (Anthropic, 2025), a proprietary model widely recognized for its strong performance in multi-turn reasoning tasks. We also conduct a comprehensive assessment of the DeepSeek-R1 family (Guo et al., 2025), including the full-scale DeepSeek-R1 and its distilled Qwen-1.5-based variants at 1.5B, 7B, 14B, and 32B scales, all explicitly optimized for efficient multi-step reasoning. Additionally, we evaluate Qwen3-235B-A22B (Team, 2025b) and Qwq-Plus (Team, 2024), both equipped with reflection and alignment mechanisms to support long-context inference. To further explore model behavior under extended reasoning conditions, we include Ernie-X1-Turbo-32K (Team, 2025a), optimized for long input sequences, along with Grok-3-Mini-Beta (xAI, 2025) and GLM-Z1-Air (GLM et al., 2024).

**Implementation Details** Throughout the evaluation process, we initially employed the tested LRMs to generate responses to the entries from Think-Bench. All other model hyperparameters followed default settings unless otherwise specified. Subsequently, Claude 3.7 Sonnet was utilized to analyze the reasoning steps and underlying thinking processes of these responses. The detailed prompt used for the analysis with Claude 3.7 Sonnet is provided in Appendix I. Finally, we computed our proposed evaluation metrics to assess the thinking efficiency and reliability of the tested LRMs.

### 4.2 QUANTITATIVE RESULTS

We conduct a comprehensive evaluation of LRMs using our proposed Think-Bench. The main results are presented in Tables 2 and Tables 3. We begin with an analysis of the overall performance, followed by an in-depth discussion of the key findings.

Table 3: Evaluation Results of CoT and Efficiency in Think-Bench Classified by Difficulty Levels. Best performance in **bold**.

| Model name | Recall | | Precision | | Reflection Quality | | Tokens | | Efficiency | |
|---|---|---|---|---|---|---|---|---|---|---|
| | Simple | Difficult | Simple | Difficult | Simple | Difficult | Simple | Difficult | Simple | Difficult |
| Claude-3.7-sonnet | 88.49% | 74.05% | 92.94% | 79.56% | 92.95% | 90.29% | **673.24** | **1216.01** | 0.52 | 0.47 |
| Deepseek-r1-distill-qwen-1.5b | 55.74% | 38.42% | 69.35% | 49.84% | 73.22% | 68.55% | 2149.94 | 5325.97 | 0.40 | 0.34 |
| Deepseek-r1-distill-qwen-7b | 70.75% | 51.30% | 88.84% | 71.06% | 90.09% | 83.64% | 1575.27 | 4059.64 | 0.51 | 0.51 |
| Deepseek-r1-distill-qwen-14b | 72.99% | 55.31% | 92.55% | 74.93% | 91.43% | 85.07% | 1514.53 | 3886.05 | 0.52 | 0.53 |
| Deepseek-r1-distill-qwen-32b | 69.88% | 57.39% | 85.79% | 68.75% | 84.99% | 78.91% | 2074.32 | 4941.47 | 0.49 | 0.50 |
| Deepseek-r1 | 88.80% | 72.77% | **95.54%** | 81.09% | 95.15% | 90.00% | 2058.35 | 5539.63 | 0.46 | 0.52 |
| Ernie-x1-turbo-32k | 90.26% | 73.76% | 95.44% | **81.88%** | 95.00% | 89.09% | 2679.32 | 6713.91 | 0.43 | 0.51 |
| Grok-3-mini-beta | 89.09% | 74.01% | 93.47% | 79.51% | 93.17% | 88.78% | 1242.27 | 2543.25 | **0.60** | **0.63** |
| Qwen3-235b-a22b | **92.87%** | **78.70%** | 95.33% | 78.57% | **96.29%** | **91.43%** | 2818.69 | 7128.80 | 0.42 | 0.50 |
| Qwq-plus | 90.04% | 70.72% | 94.89% | 75.23% | 94.60% | 87.42% | 3289.45 | 8197.99 | 0.41 | 0.48 |
| GLM-Z1-Air | 88.69% | 71.60% | 93.31% | 72.99% | 94.79% | 86.55% | 1931.36 | 5433.63 | 0.45 | 0.49 |

**Overall Performance** To comprehensively evaluate the performance of LRMs, we report results across two key dimensions: efficiency and CoT quality, including our proposed efficiency metrics, recall, precision and accuracy, as shown in Table 2. Our analysis shows that while there exists a consistent trade-off between token usage and reasoning performance, different models exhibit significant variability in their inference behaviors.

In terms of efficiency, Grok-3-mini-beta achieves the highest score of 61.69%, followed by Deepseek-r1-distill-qwen-32b at 52.62% and Deepseek-r1-distill-qwen-14b at 50.70%, indicating a more economical use of tokens to reach correct answers. In contrast, larger models such as Qwen3-235b-a22b and Qwq-plus exhibit lower efficiency, scoring 46.14% and 44.58% respectively. This

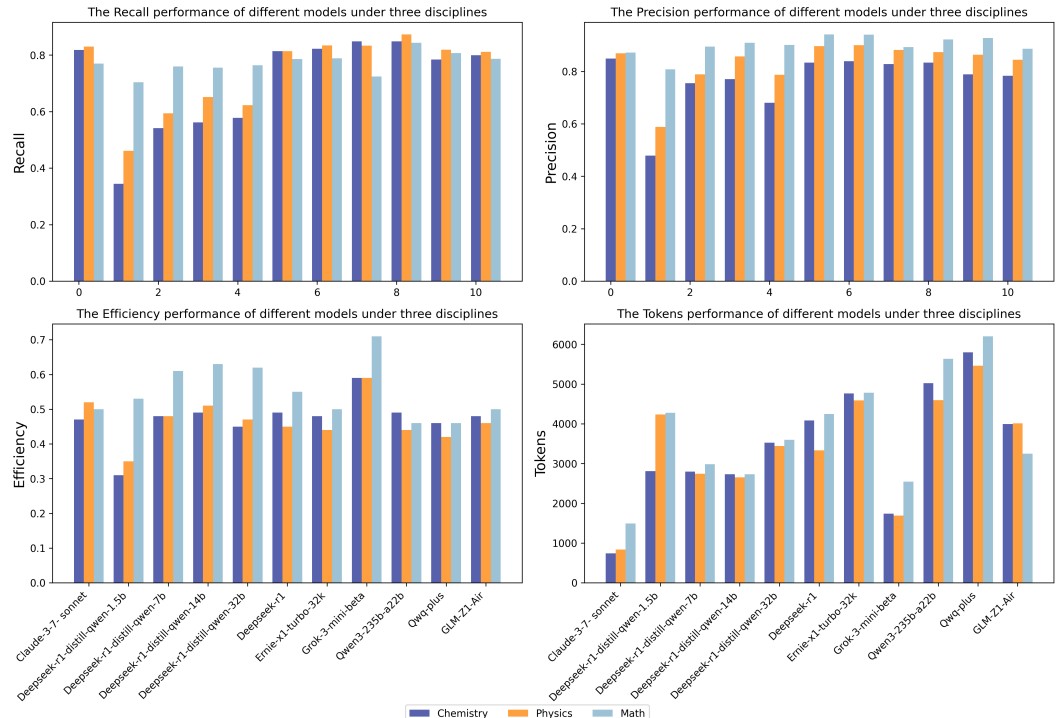

Figure 6: Comparative Performance of Models in Chemistry, Physics, and Math.

decrease in efficiency is attributed to their prolonged reasoning chains, despite having strong CoT quality.

Regarding CoT quality, Qwen3-235b-a22b and Ernie-x1-turbo-32k stand out by achieving the highest reflection quality scores, with values of 92.16% and 90.97%, respectively. They also demonstrate top-tier precision at 86.97% and 88.67%, and recall rates of 85.80% and 82.03%. These impressive results highlight the advantages of large-scale models with reflection-enhanced reasoning capabilities, which not only lead to accurate conclusions but also enable reliable verification and correction processes. In contrast, smaller distilled models, such as Deepseek-r1-distill-qwen-1.5b, perform poorly across all quality metrics, particularly in precision (59.61%) and recall (47.10%).

An important behavioral indicator is Thought Num, reflecting how often the model switches or reconsiders its CoT. Qwq-plus shows the highest value (22.63), indicating frequent reflective iterations. However, such reflections don't always lead to better performance and may reduce efficiency. In contrast, models like Claude 3.7 Sonnet and Grok-3-mini-beta maintain very low Thought Num values (0.28 and 0.38) while still achieving a balanced and high-quality reasoning process.

Regarding token consumption, Qwq-plus and Qwen3-235b-a22b each use over 4,900 tokens per response, with a substantial portion from reflection (3,091.64 and 2,520.76, respectively). This suggests tendencies toward overthinking. In contrast, Claude-3-7-Sonnet completes its reasoning in under 1,000 tokens, demonstrating concise and effective inference with minimal redundancy.

**Evaluation and Analysis Based on Difficulty Levels of Questions**   The results in Table 3 show that most LRMs demonstrate significantly lower average efficiency on simple questions than on difficult ones. This suggests that when faced with simple questions, these models tend to overthink and generate unnecessary reasoning chains. In contrast, for high difficulty questions, the models focus more effectively, eliminating redundant reasoning steps and improving efficiency. Additionally, token consumption for difficult questions is consistently higher than for simple ones, due to the extra inferential steps needed to tackle complex problems. As reflection quality, recall, and precision all decline slightly as task difficulty increases, this illustrates that while difficult questions require more computational resources, they present greater challenges to the model's reasoning capabilities.

**Evaluation and Analysis Based on Different Subjects**  As shown in Figure 6, the multi-disciplinary evaluation in Think-Bench reveals notable performance differences across chemistry, physics, and mathematics.  Mathematical tasks generally lead to higher token consumption and lower reasoning efficiency, even for strong models, suggesting a reliance on lengthy CoTs and structured outputs.  In contrast, chemistry and physics tasks typically exhibit better efficiency and lower token usage.

Regarding CoT quality evaluation, the recall and precision generally show a positive correlation in various disciplinary tasks within Think-Bench, but there are also obvious structural differences. Specifically, in chemistry and physics tasks, the precision rate of the model is often significantly better than the recall rate.  This phenomenon reflects that the current LRMs' strategy in generating answers for reasoning questions is relatively conservative, such that it is more inclined to output answers with high confidence.

### 4.3 ERROR ANALYSIS

**LRM response error analysis**  The LRM's reasoning errors typically arise from misinterpretation of key problem conditions, improper selection and application of relevant theorems or formulas. Such issues often lead to distorted variable constraints, flawed inferential logic, and incorrect quantitative calculations, ultimately resulting in deviated final results.  As illustrated in Figure 11 in Appendix D.3, a typical manifestation of such errors is the misinterpretation of multi-scenario constraints inherent in the problem, failure to comprehensively consider all valid condition boundaries, and consequent one-sided logical derivation.

**Other error analysis**  During the evaluation experiment, we observed that some models, particularly the distillation models from the DeepSeek series and the ERNIE-X1-Turbo-32K, exhibited an unusual issue of generating empty outputs.  This issue primarily manifested in the model generating only intermediate reasoning content without producing a final answer for certain questions. See Figure 10 in Appendix D.3 for a concrete example.  Potential causes include limitations in their inference mechanisms, context processing capabilities, or deployment implementations.  Given that such outputs fail to meet the basic requirement of providing a complete final answer, we directly marked these samples as model response errors to maintain the rigor and consistency of the evaluation.

## 5 CONCLUSION

This paper presents the Think-Bench dataset, a benchmark designed to systematically evaluate the reasoning efficiency and CoT quality of LRMs.  The dataset consists of tasks from three disciplines: mathematics, physics, and chemistry. Each task is provided at two difficulty levels: Simple and Difficult.  Evaluation is conducted using nine metrics, including six efficiency indicators, two CoT quality measures and accuracy.  To verify the effectiveness of Think-Bench and to assess the reasoning efficiency and CoT quality of mainstream LRMs, we conduct a comprehensive evaluation of 11 representative models. Experimental results show that most models exhibit overthinking behaviors on simple questions, generating excessive reasoning tokens and leading to unnecessary computational overhead.  This study not only highlights the limitations of current LRMs in their use of computational resources in inference time, but also offers insights that may inspire future research, including designing dynamic reasoning pathways, early exit mechanisms, and enhancing adaptability across disciplines.

### REPRODUCIBILITY STATEMENT

We have made every effort to ensure the reproducibility of our results.  The design of the Think-Bench benchmark, including dataset curation, annotation pipeline, and evaluation metrics, is described in detail in Section 2, Section 3 and Appendix I. To facilitate replication of our experiments, we provide an anonymous repository containing the full source code and processed datasets, available at `https://anonymous.4open.science/r/Think-Bench-anony-6866`.  The repository includes scripts for data preprocessing, evaluation, and reproducing all reported results. Together, these resources are intended to allow independent researchers to fully reproduce and extend our findings.

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

## A RELATED WORK

In recent years, evaluating the reasoning capabilities of LLMs has become a pivotal research focus within the field of natural language processing(Chang et al., 2024). Existing evaluation methodologies can be broadly categorized into two approaches: outcome-oriented and process-oriented assessments.

Outcome-oriented evaluations primarily emphasize the accuracy of the model's final output. Prominent benchmarks in this category include SuperGLUE (Wang et al., 2019), MMLU (Hendrycks et al., 2020b), and BIG-bench (Suzgun et al., 2022). These benchmarks encompass a wide array of tasks, ranging from language comprehension to domain-specific question answering, thereby standardizing the performance assessment of LLMs. However, such methods often overlook the interpretability and rationality of the model's reasoning process, particularly in complex problem-solving scenarios where the significance of intermediate steps is substantially undervalued.

To address these limitations, process-oriented evaluation methodologies have been introduced(Zheng et al., 2024; Jiang et al., 2025). The CoT reasoning framework (Wei et al., 2022) exemplifies this approach by explicitly guiding models to generate intermediate reasoning steps, thereby enhancing performance in mathematical and logical tasks. Subsequent studies, such as Auto-CoT (Zhang et al., 2022), Tree-of-Thought (Yao et al., 2023a), and ReAct (Yao et al., 2023b), have further augmented the flexibility and diversity of reasoning pathways.

Furthermore, the evaluation of multidisciplinary reasoning capabilities has become a focal point in current research. Researchers have developed various assessment benchmarks and methodologies

to comprehensively measure the reasoning abilities of LLMs across different academic disciplines. For instance, the Advanced Reasoning Benchmark (ARB) is a comprehensive reasoning benchmark that spans multiple domains, including mathematics, physics, biology, chemistry, and law, designed to evaluate the performance of LLMs in complex reasoning tasks (Sawada et al., 2023). Multi-LogiEval is a dataset that provides an integrated evaluation of LLMs' multi-step logical reasoning abilities, covering multiple types of logic such as propositional logic, first-order logic, and non-monotonic logic (Patel et al., 2024).

Additionally, large reasoning models tend to exhibit overthinking behavior during chain-of-thought reasoning, where excessively long and unnecessary reasoning steps are generated even for simple or ill-posed problems (Sui et al., 2025; Chen et al., 2024; Fan et al., 2025; Pu et al., 2025). This phenomenon is often attributed to the models' lack of proper termination mechanisms and insufficient confidence estimation, leading to inefficient inference and degraded accuracy. It has been observed that the issue becomes more pronounced when essential premises are missing from the input (Fan et al., 2025). To address this, several approaches have been proposed, including the introduction of new reasoning efficiency metrics and self-training strategies that encourage concise reasoning (Chen et al., 2024), as well as dynamic early-exit mechanisms that halt inference when sufficient confidence is reached (Yang et al., 2025b). Additionally, path scoring methods have been developed to prefer less redundant reasoning paths, thereby improving performance while reducing computational cost (Cuadron et al., 2025).

## B MORE EXPERIMENTAL RESULTS

The comparative analysis of Tables 4 and Tables 5 highlights the trade-offs between reasoning quality and efficiency across various disciplines. In mathematics, top-performing models, such as Qwen3-235b-a22b and Qwq-plus, achieve high quality of reflection and precision but require a large number of tokens. In contrast, Grok-3-mini-beta strikes a balance between conciseness and accuracy, achieving a precision of 88.2% in physics and a recall of 84.9% in chemistry while using fewer tokens. Furthermore, smaller distilled variants, like Deepseek-r1-distill-qwen-1.5b, demonstrate significant limitations in domain-specific reasoning, particularly in physics and chemistry, where both recall and precision fall below 50%.

Table 4: **Comparative Performance of Models in Different Category.** C-3.7-sonnet: claude 3.7 sonnet; Ds-r1-distill-qwen-1.5b: deepseek-r1-distill-qwen-1.5b; Ds-r1-distill-qwen-7b: deepseek-r1-distill-qwen-14b; Ds-r1-distill-qwen-14b: deepseek-r1-distill-qwen-32b; Ds-r1-distill-qwen-32b: deepseek-r1-distill-qwen-7b; Ds-r1: deepseek-reasoner; Es-x1-turbo-32k: ernie-x1-turbo-32k; G-3-mini-beta: grok-3-mini-beta; Q3-235b-a22b: qwen3-235b-a22b.

| Model name | Recall | | | Precision | | | Reflection Quality | | | Tokens | | |
|---|---|---|---|---|---|---|---|---|---|---|---|---|
| | Chemistry | Physics | Math | Chemistry | Physics | Math | Chemistry | Physics | Math | Chemistry | Physics | Math |
| C-3.7-sonnet | 81.83% | 82.97% | 76.97% | 84.92% | 86.94% | 87.16% | 87.89% | 90.39% | 95.25% | 744.21 | 836.63 | 1494.49 |
| Ds-r1-distill-qwen-1.5b | 34.43% | 46.11% | 70.37% | 47.91% | 58.86% | 80.78% | 59.37% | 70.18% | 89.20% | 2807.87 | 4236.72 | 4279.11 |
| Ds-r1-distill-qwen-7b | 54.11% | 59.42% | 75.95% | 75.55% | 78.93% | 89.49% | 80.16% | 87.28% | 95.58% | 2800.44 | 2742.55 | 2984.15 |
| Ds-r1-distill-qwen-14b | 56.21% | 65.16% | 75.54% | 77.07% | 85.76% | 90.95% | 80.56% | 90.89% | 96.36% | 2729.47 | 2654.41 | 2731.53 |
| Ds-r1-distill-qwen-32b | 57.75% | 62.24% | 76.39% | 68.02% | 78.67% | 90.09% | 72.12% | 83.86% | 94.46% | 3525.06 | 3442.9 | 3595.26 |
| Ds-r1 | 81.39% | 81.41% | 78.59% | 83.34% | 89.63% | 94.09% | 87.79% | 93.32% | 98.05% | 4082.93 | 3331.74 | 4245.25 |
| Es-x1-turbo-32k | 82.25% | 83.44% | 78.82% | 83.92% | 90.00% | 94.00% | 87.95% | 92.75% | 97.00% | 4762.48 | 4588.33 | 4783.4 |
| G-3-mini-beta | 84.86% | 83.34% | 72.45% | 82.79% | 88.22% | 89.30% | 86.04% | 91.69% | 96.53% | 1742.1 | 1690.57 | 2546.22 |
| Q3-235b-a22b | 84.88% | 87.27% | 84.37% | 83.39% | 87.36% | 92.18% | 90.29% | 94.55% | 98.05% | 5022.55 | 4592.91 | 5636.6 |
| Qwq-plus | 78.38% | 81.92% | 80.73% | 78.87% | 86.42% | 92.80% | 84.76% | 92.29% | 98.11% | 5795.2 | 5460.61 | 6202.25 |
| Glm-z1-air | 79.93% | 81.09% | 78.66% | 78.37% | 84.49% | 88.66% | 84.51% | 91.84% | 96.97% | 3992.9 | 4013.77 | 3250.19 |

As shown in Table 6. On simple tasks, Grok-3-mini-beta demonstrates efficient and focused reasoning, producing only 0.54 thoughts and consuming 1,574.26 tokens. In contrast, when tackling difficult questions, larger models such as Qwen3-235b-a22b and Ernie-x1-turbo-32k generate over 3,600 tokens on average while achieving high reflection quality, reaching 92.16% and 94.00% respectively. However, this increase in quality comes with reduced efficiency. For example, Qwq-plus achieves only 44.58% efficiency due to its high reflection token count after the answer, totalling 4,226.88 tokens.

Table 5: **Comparative Performance of Models in Different Category.** C-3.7-sonnet: claude 3.7 sonnet; Ds-r1-distill-qwen-1.5b: deepseek-r1-distill-qwen-1.5b; Ds-r1-distill-qwen-7b: deepseek-r1-distill-qwen-14b; Ds-r1-distill-qwen-14b: deepseek-r1-distill-qwen-32b; Ds-r1-distill-qwen-32b: deepseek-r1-distill-qwen-7b; Ds-r1: deepseek-reasoner; Es-x1-turbo-32k: ernie-x1-turbo-32k; G-3-mini-beta: grok-3-mini-beta; Q3-235b-a22b: qwen3-235b-a22b.

| Model name | Thought Num | | | Efficiency | | | Useful Tokens | | | Reflection Tokens | | |
| --- | --- | --- | --- | --- | --- | --- | --- | --- | --- | --- | --- | --- |
| | Chemistry | Physics | Math | Chemistry | Physics | Math | Chemistry | Physics | Math | Chemistry | Physics | Math |
| C-3.7-sonnet | 0.16 | 0.23 | 0.58 | 0.47 | 0.52 | 0.5 | 339.85 | 411.6 | 695.93 | 404.35 | 425.03 | 798.55 |
| Ds-r1-distill-qwen-1.5b | 8.2 | 8.65 | 6.35 | 0.31 | 0.35 | 0.53 | 842.08 | 1243.98 | 2033.23 | 1965.79 | 2992.74 | 2245.88 |
| Ds-r1-distill-qwen-7b | 9.76 | 6.13 | 4.3 | 0.48 | 0.48 | 0.61 | 1367.99 | 1243.05 | 1831.2 | 1432.45 | 1499.5 | 1152.94 |
| Ds-r1-distill-qwen-14b | 8.24 | 5.75 | 4 | 0.49 | 0.51 | 0.63 | 1270.29 | 1226.93 | 1745.42 | 1459.18 | 1427.48 | 986.11 |
| Ds-r1-distill-qwen-32b | 12.49 | 8.43 | 6.25 | 0.45 | 0.47 | 0.62 | 1535.14 | 1480.06 | 2147.1 | 1989.91 | 1962.84 | 1448.16 |
| Ds-r1 | 13.13 | 6.96 | 6.97 | 0.49 | 0.45 | 0.55 | 2087.08 | 1521.81 | 2404.29 | 1995.86 | 1809.93 | 1840.97 |
| Es-x1-turbo-32k | 17.14 | 11.15 | 8.62 | 0.48 | 0.44 | 0.5 | 2315.51 | 2031.88 | 2444.63 | 2446.98 | 2556.44 | 2338.77 |
| G-3-mini-beta | 0.45 | 0.24 | 0.52 | 0.59 | 0.59 | 0.71 | 1049.76 | 985.93 | 1738.09 | 692.34 | 704.65 | 808.13 |
| Q3-235b-a22b | 16.95 | 11.21 | 11.62 | 0.49 | 0.44 | 0.46 | 2547.58 | 2087.76 | 3007.54 | 2474.97 | 2505.15 | 2629.06 |
| Qwq-plus | 30.98 | 18.24 | 17.44 | 0.46 | 0.42 | 0.46 | 2664.93 | 2351.91 | 3209.83 | 3130.26 | 3108.69 | 2992.42 |
| Glm-z1-air | 14.11 | 6.99 | 7.61 | 0.48 | 0.46 | 0.50 | 1887.16 | 1483.43 | 2174.12 | 2105.78 | 1766.76 | 1839.66 |

Table 6: **Effect of question difficulty on other efficiency measures.**

| Model name | Thought Num | | Useful Tokens | | Reflection Tokens | |
| --- | --- | --- | --- | --- | --- | --- |
| | Simple | Difficult | Simple | Difficult | Simple | Difficult |
| Claude-3.7-sonnet | 0.17 | 0.39 | 340.05 | 553.55 | 333.19 | 662.46 |
| Deepseek-r1-distill-qwen-1.5b | 4.10 | 11.91 | 818.28 | 1720.40 | 1331.65 | 3605.56 |
| Deepseek-r1-distill-qwen-7b | 3.05 | 11.04 | 825.64 | 2003.10 | 749.63 | 2056.54 |
| Deepseek-r1-distill-qwen-14b | 2.86 | 9.70 | 777.32 | 1931.06 | 737.21 | 1954.99 |
| Deepseek-r1-distill-qwen-32b | 4.68 | 14.18 | 960.86 | 2325.93 | 1113.45 | 2615.53 |
| Deepseek-r1 | 4.26 | 14.11 | 950.72 | 2877.72 | 1107.63 | 2661.91 |
| Ernie-x1-turbo-32k | 6.75 | 18.78 | 1144.62 | 3302.73 | 1534.70 | 3411.17 |
| Grok-3-mini-beta | 0.22 | 0.54 | 765.60 | 1574.26 | 476.67 | 968.99 |
| Qwen3-235b-a22b | 7.93 | 18.80 | 1245.45 | 3656.38 | 1573.24 | 3472.42 |
| Qwq-plus | 11.09 | 34.21 | 1328.11 | 3971.12 | 1961.34 | 4226.88 |
| Glm-z1-air | 4.27 | 15.36 | 876.64 | 2677.43 | 1054.72 | 2756.20 |

Table 7: The results of the thinking process model are not provided.

| Model | Recall | Precision | Accuracy |
| --- | --- | --- | --- |
| Don't provide thinking content | | | |
| GPT-5 | 70.65% | 95.41% | 93.16% |
| o4-mini | 73.25% | 92.76% | 95.42% |
| Gemini2.5-pro | **88.46%** | **95.69%** | **97.16%** |
| Provide thinking content | | | |
| Claude-3.7-sonnet | 81.29% | 86.26% | 93.89% |
| Deepseek-r1-distill-qwen-1.5b | 47.10% | 59.61% | 60.58% |
| Deepseek-r1-distill-qwen-7b | 63.65% | 77.29% | 81.53% |
| Deepseek-r1-distill-qwen-14b | 61.04% | 79.97% | 80.65% |
| Deepseek-r1-distill-qwen-32b | 64.17% | 83.76% | 83.71% |
| Deepseek-r1 | 80.80% | 88.33% | 93.82% |
| Ernie-x1-turbo-32k | 82.03% | **88.67%** | 92.36% |
| Grok-3-mini-beta | 81.56% | 86.51% | 93.96% |
| Qwen3-235b-a22b | **85.80%** | 86.97% | **94.98%** |
| Qwq-plus | 80.40% | 85.08% | 90.76% |
| Glm-z1-air | 80.16% | 83.18% | 91.92% |

As shown in Table 7, we compare the Recall, Precision, and Accuracy of models for which the platform does not provide intermediate thinking content. Overall, Gemini 2.5 Pro achieves the best performance across all three metrics, demonstrating stable and reliable reasoning capability. In contrast, GPT-5 and o4-mini exhibit abnormally low Recall scores. Further analysis reveals that this phenomenon does not stem from insufficient reasoning ability. Instead, for some questions, these models return only the final answer without any reasoning steps that could be aligned with the annotated key steps. Consequently, their matched-step counts are systematically underestimated, leading to depressed Recall values.

To explore the influence of output length, we conduct a token-budget study on GPT-5. Table 8 shows that increasing the token budget from 3000 to unlimited improves Accuracy from 75.85% to 93.16%. This demonstrates that limiting the token budget affects the completeness of the model's outputs and consequently its measurable performance.

Table 8: Impact of token budget on Accuracy, taking GPT-5 as an example

| Token Budget | Accuracy |
|---|---|
| 3000 | 75.85% |
| 5000 | 84.87% |
| No limited | 93.16% |

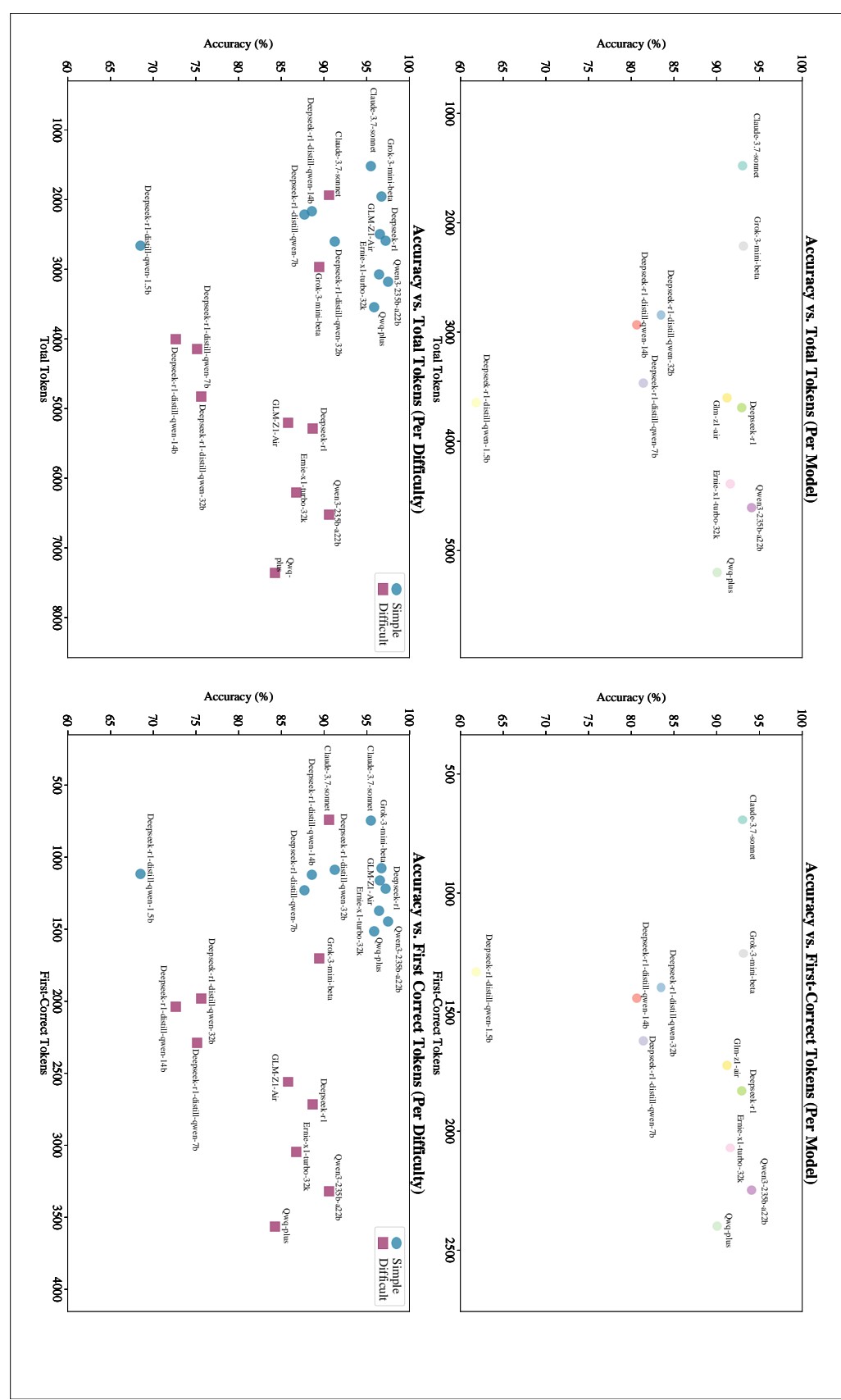

Figure 7: Relationship Diagram of Accuracy vs. Total Tokens and Accuracy vs. First Correct Tokens Across Different Models and Difficulty Levels.

## C    IN-DEPTH ANALYSIS OF THE RESULTS

We conduct an in-depth analysis of the experimental results by integrating the specific mechanisms of model training and decoding design. First, Grok-3-mini-beta, one of the most efficient models in the experiment, exhibits a small number of Thought Num, low Reflection Tokens, and the highest Efficiency. The core reason for this phenomenon lies in the fact that the Grok-3 series models extensively adopt reinforcement learning or targeted strategy optimization during training to refine reasoning paths, enabling the model to acquire the ability to achieve goals or complete necessary self-verification through fewer steps. This further reduces redundant generation and directly improves the ratio of First-Correct Tokens to Total-Tokens.

In contrast, models with better performance in CoT quality metric, such as Qwen3-235b-a22b and Ernie-x1-turbo-32k, tend to adopt or allow prolonged extended thinking or multi-path sampling strategies. These models generate multiple or longer reasoning chains through test-time scaling, self-consistency sampling, or explicit reflection or verification mechanisms, and integrate answers via majority voting to enhance the stability and accuracy of results. Relevant existing studies have confirmed that self-consistency sampling can significantly improve the correctness of final answers, but it comes at the cost of generating more tokens, thereby reducing Efficiency.

Furthermore, distilled or pruned variant models demonstrate low Recall and Precision in the experiment, which is consistent with the inherent characteristic of the distillation process. Although distillation technology helps improve model throughput and the explicit conciseness of outputs, it is prone to losing the completeness of intermediate reasoning steps or failing to fully retain alignable key reasoning components, ultimately leading to a decline in CoT quality. The experimental data table also shows that the efficiency of small distilled models is not always superior, while their quality decreases significantly. Finally, some models extend reasoning chains when necessary through extended thinking modes or controllable thinking budget mechanisms. This design explains why Claude can maintain high reflection quality while using a relatively small number of total tokens: the model implements a scheduling mechanism of slowing down for verification when necessary and terminating reasoning as early as possible when unnecessary, thus achieving a balanced advantage between efficiency and quality.

Figure 7 intuitively illustrates a prominent token complexity phenomenon (Lee et al., 2025). First, with the increase in problem difficulty and the model's tendency toward in-depth verification, the total number of tokens exhibits a non-linear growth. Specifically, a large number of additional reflection or verification steps do not necessarily lead to a proportional improvement in accuracy. This trend is observed across multiple large language models, with accuracy plateauing in the high token consumption range. Second, when analyzing with First Correct Tokens as the horizontal axis, the curves of different models demonstrate the critical role of early correct convergence in reasoning efficiency. When the First Correct Tokens are low and the Efficiency is high, the model can significantly reduce the total computational cost while maintaining accuracy. In contrast, scenarios with both high First Correct Tokens and high Reflection Tokens often correspond to high accuracy but extremely high token complexity, a situation that is more prone to inducing overthinking especially for simple problems.

## D    MORE QUALITATIVE EXAMPLES

### D.1    EXAMPLE OF COMPUTATIONAL EFFICIENCY

In Figure 8, we illustrate an example of efficient reasoning by an LRM.

**Question** Three pencils and a jumbo eraser cost $\$1.24$. Five pencils and a jumbo eraser cost $\$1.82$. No prices include tax. In cents, what is the cost of a pencil?

**Answer** 29

🐋 **Deepseek R1**

**Solution 1** Okay, so I need to figure out how much a pencil costs, right? The problem says three pencils and a jumbo eraser cost $1.24, and five pencils and a jumbo eraser cost $1.82. They want the cost of a pencil in cents. Hmm, let me think.\nFirst, maybe I should set up some equations. Let me call the cost of one pencil \"p\" and the cost of a jumbo eraser \"e\". Both are in dollars for now, but since the answer needs to be in cents, I'll convert later.\n\nSo, the first equation would be $3p + e = 1.24$. That's three pencils plus an eraser equals $1.24. The second equation is $5p + e = 1.82$. Five pencils and the same eraser cost $1.82.\n\nHmm, okay, so I have two equations here with two variables. That means I can solve them using substitution or elimination. Let's see. If I subtract the first equation from the second, maybe that will eliminate the eraser?\n\nLet me try that. So, subtract $(3p + e)$ from $(5p + e)$:\n\n$(5p + e) - (3p + e) = 1.82 - 1.24$\n\nSimplifying the left side: $5p - 3p + e - e = 2p$. The right side is $1.82 - 1.24$, which is $0.58$. So, $2p = 0.58$. Then, divide both sides by 2: $p = 0.58 / 2 = 0.29$. Wait, that's in dollars. So, converting to cents, that's 29 cents per pencil.

**Solution 2** Let me check if that makes sense.\n\nIf a pencil is 29 cents, then three pencils are $3 * 29 = 87$ cents. Then, the eraser would be $1.24 - 0.87 = 0.37$ dollars, which is 37 cents. Then, check the second equation: five pencils would be $5 * 29 = 145$ cents, which is $1.45. Adding the eraser, $1.45 + 0.37 = 1.82$, which matches the given amount. So, that works out.

**Solution 3** Wait, but let me just confirm my steps again to make sure I didn't make a mistake. Starting with the two equations:\n\n1) $3p + e = 124$ cents (since $1.24 is 124 cents)\n2) $5p + e = 182$ cents\n\nSubtract equation 1 from equation 2:\n\n$(5p + e) - (3p + e) = 182 - 124$\n\nWhich simplifies to:\n\n$2p = 58$\n\nSo, $p = 58 / 2 = 29$ cents. Yep, same result. So, that seems right.

**Solution 4** Alternatively, if I didn't convert to cents initially, the equations would be in dollars:\n\n$3p + e = 1.24$\n$5p + e = 1.82$\n\nSubtract first equation from the second:\n\n$2p = 0.58$ → $p = 0.29$ dollars, which is 29 cents. Same answer. Either way works.\n\nSo, the cost of a pencil is 29 cents. Both methods confirm that. The eraser ends up being 37 cents, which checks out in both equations. So, I think that's the correct answer. I don't see any mistakes in the calculations, so I'm pretty confident.\n\n**Final Answer**\nThe cost of a pencil is \\boxed{29} cents.

**Efficiency = 278/819=33.9%**

Figure 8: **Examples of Efficiency Evaluation.**

## D.2 REFLECTION QUALITY EXAMPLE

Figure 9 presents the judgment examples of self-reflection in the reasoning chains generated by the model. The judgment process strictly adheres to the Reflection Quality prompt provided in the Ap-

pendix 6: first, candidate reflection steps are identified through several trigger phrases; subsequently, each reflection is labeled as "Correct" or "Wrong" in accordance with explicit evaluation criteria. The specific criteria are defined as follows: a reflection is deemed valid if it explicitly identifies and corrects errors in previous reasoning by introducing new analytical perspectives or evidence, or draws a confirmatory conclusion consistent with the annotated ground truth through additional reasoning. In contrast, a reflection is classified as invalid if it merely repeats existing content without providing new information, derives conclusions inconsistent with the ground truth, proposes but fails to implement new analytical methods, or falls into other types of errors. During the experiment, the evaluation process strictly retains the order and content of the original reflections without adding extra explanations, and outputs the judgment results in a structured format as required, which ensures the reliability of the experimental results.

> Let a, b, c, d be distinct complex numbers such that $|a| = |b| = |c| = |d| = 1$ and $a + b + c + d = 0$.
> Find the maximum value of $|(a + b)(a + c)(a + d)(b + c)(b + d)(c + d)|$.

**LRM**

> Given $|a|=|b|=|c|=|d|=1$, we have $a\overline{a}=b\overline{b}=c\overline{c}=d\overline{d}=1$. From $a+b+c+d=0$, it follows that $b+c+d=-a$. Expanding $(a+b)(a+c)(a+d)$ and substituting $b+c+d=-a$, we simplify it to $a(bc+bd+cd)+bcd$. Since modulus reflects magnitude, it seems reasonable that $|x+y|=|x|+|y|$, so: $|a(bc+bd+cd)+bcd|=|bc+bd+cd|+1$. With $|bc|=|bd|=|cd|=1$, adding their moduli gives $|bc+bd+cd|=3$, so this product's modulus is 4. Applying the same logic to the other terms, the total product's modulus should max out at $4\times4\times4=64$. Wait, let's test with concrete values: take $a=1$, $b=-1$, $c=i$, $d=-i$ (they're distinct, have modulus 1, and sum to 0). Calculating the product gives 0—this conflicts with the earlier result.

> Maybe this set is just an exception? I try another combination: $a=1$, $b=e^{i\pi/3}$, $c=e^{i2\pi/3}$, then $d=-(1+e^{i\pi/3}+e^{i2\pi/3})=0$, which doesn't satisfy $|d|=1$.

> Hmm, why can't I find a valid set with a positive product? Wait, let's revisit the key equation: $a+b+c+d=0$ means $a+b=-(c+d)$, so $|a+b|=|c+d|$. If $a=-b$ (they're distinct, so this works), then $|a+b|=0$, which forces $|c+d|=0$, so $c=-d$. Maybe this is how all valid sets are structured—no wonder the test case gave 0.
> If every valid set has $a=-b$ and $c=-d$, then $a+b=0$ and $c+d=0$ will always be factors in the product, making the total product 0. Since all attempts to find a valid set with a positive product modulus failed, 0 must be the maximum value. Hence, the answer is $\boxed{0}$.

Figure 9: Examples of Reflection Quality. Red is invalid content reflection, green is effective reflection.

## D.3 ERROR EXAMPLE

We present an example of an LRM output error in Figure 10 and a reasoning error in Figure 11.

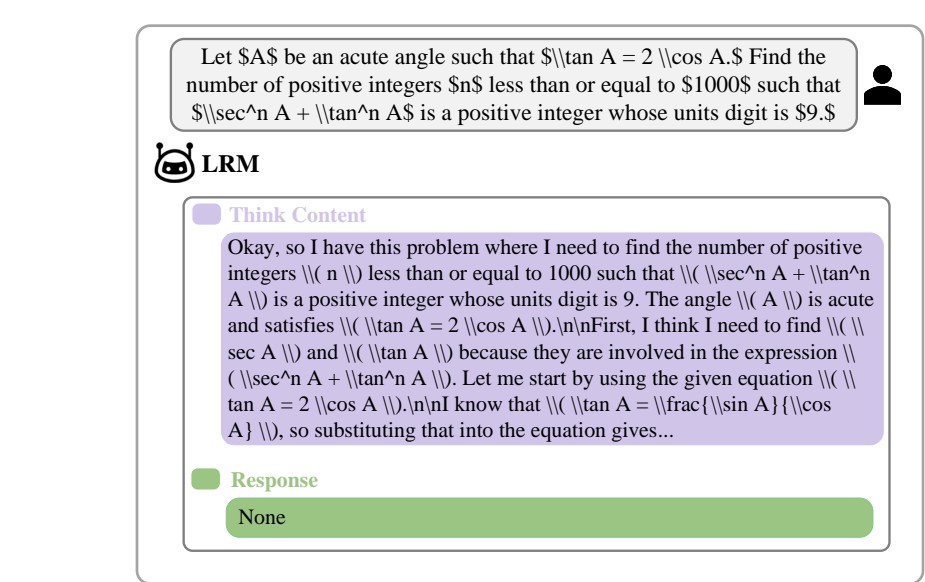

Figure 10: Failure Case Example in the Deepseek-r1-distill-qwen-32b Answering Process.

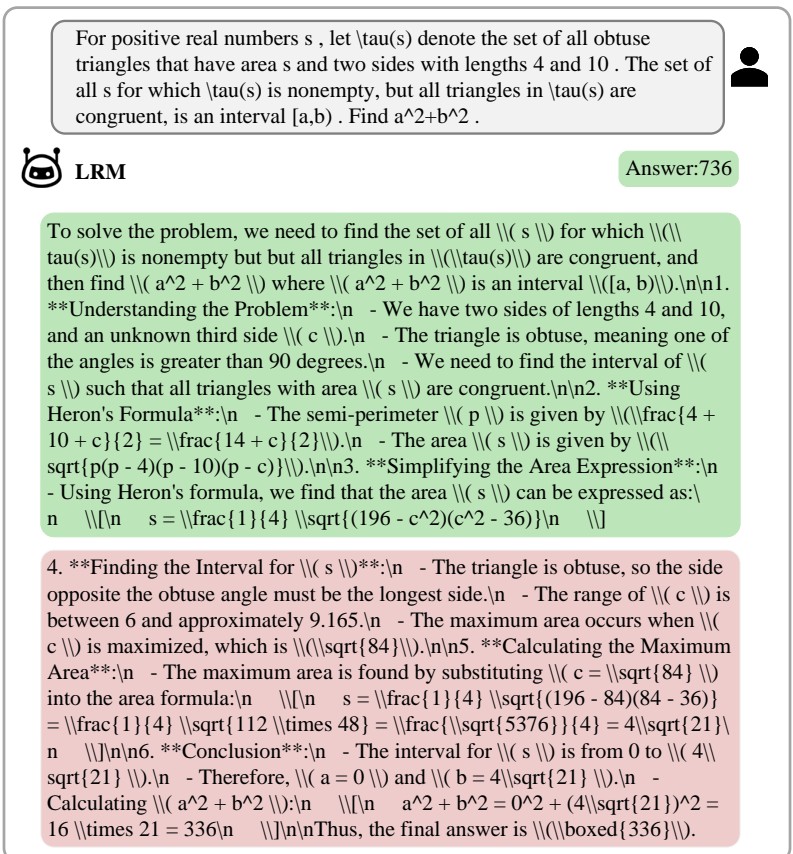

Figure 11: Problem Misinterpretation Error Case (Deepseek-r1-distill-qwen-32b).

# E    ANNOTATION CONSISTENCY AND JUDGE MODEL RELIABILITY EVALUATION

To ensure the validity and robustness of the Think-Bench benchmark, we conduct two critical evaluations: the consistency of human annotators during key step annotation, and the reliability of Claude 3.7 Sonnet as the judge model for assessing reasoning processes. This section details the experimental design, results, and analysis of these evaluations.

## E.1    ANNOTATION CONSISTENCY EVALUATION

We recruit two graduate-level annotators to participate in the annotation consistency test. All annotators receive unified training on the annotation guidelines (detailed in Section 2.3), including the definition of key reasoning steps, annotation scope, and standard formatting requirements.

**Experimental Design**    A random sample of 300 instances is selected from Think-Bench. Each instance is annotated independently by A1 and A2. For each candidate step in an instance, both annotators label it as either "Key" or "Non-Key". After completing annotations for all 300 instances, we first count the number of instances where the two annotators agree on step labels, then use Cohen's Kappa coefficient to quantify their consistency.

**Results and Analysis**    Table 9 summarizes the agreement and disagreement counts between A1 and A2 across the 300 instances. An "Agreed" instance means A1 and A2 share identical "Key/Non-Key" labels for all steps in the instance; a "Disagreed" instance means there is at least one step where their labels differ. Among the 300 instances, 252 are marked as Agreed, and 48 are marked as Disagreed. This results in an overall raw agreement rate of 94% (282/300), indicating strong initial alignment between the two annotators, and the Cohen's Kappa coefficient for their consistency is 0.878.

Table 9: Confusion Matrix Between Two Annotators (300 Instances)

| Annotator A | Annotator B | |
| --- | --- | --- |
| | **Key** | **Non-Key** |
| Key | 120 | 10 |
| Non-Key | 8 | 162 |

## E.2    JUDGE MODEL RELIABILITY EVALUATION

To ensure the reliability of the automatic evaluation in Think-Bench, we begin by examining the reliability of the judge model. Our analysis combines both a fine-grained comparison with human annotations and a broader stability study across several candidate judge models.

We first validate judgment quality at the step level. From the Think-Bench dataset, we sample 100 representative instances and extract all annotated key reasoning steps, resulting in 968 steps in total. Each step is independently labeled as *Valid* or *Invalid* by two human annotators using the same rubric later applied to LLM judges. Claude 3.7 Sonnet is then prompted to evaluate the identical set of steps (see Appendix I). This setup allows us to directly compare Claude's decisions with the human consensus using both Cohen's Kappa and a confusion matrix.

Beyond step-level validation, we further assess the stability of multiple candidate judge models. Table 10 reports the Inter-Annotator Agreement (IAA) and Kappa scores for GPT-4o-mini, o3-mini, Qwen-2.5-72B-instruction, and Claude 3.7 Sonnet. Among these models, Claude consistently achieves the strongest alignment with human annotators, exhibiting both the highest agreement rate and the highest Kappa value. This superior consistency is the primary reason we adopt Claude 3.7 Sonnet as the judge model throughout our experiments.

**Results and Analysis** Table 11 summarize the step-level evaluation results. Across 968 reasoning steps extracted from 100 Think-Bench instances, Claude 3.7 Sonnet demonstrates a high level of alignment with human judgments. It achieves an overall agreement rate of 0.95 with human annotators and a Kappa coefficient of 0.85, indicating strong consistency between automated and human evaluations. The results suggest that Claude 3.7 Sonnet provides reliable and fine-grained assessments of individual reasoning steps, supporting its effectiveness as the judging model for Think-Bench.

### E.3 NOISE DISTURBANCE EXPERIMENT

As shown in Table 12, to assess the robustness of the judge model against real-world variations in LRM outputs, we conduct a noise disturbance experiment. We introduce controlled noise to LRM outputs through random word replacements and deletions, simulating common perturbations in generated content. We then evaluate the stability of judge model metrics under both original and noisy input conditions. The results demonstrate that Claude 3.7 Sonnet maintains remarkable stability when processing noisy LRM outputs. Its core evaluation metrics exhibit negligible fluctuations compared to the original input scenario. This resistance to minor output perturbations confirms that both Claude 3.7 Sonnet and the designed evaluation metrics exhibit robust stability, reliably maintaining consistent evaluation standards even when faced with imperfect or slightly distorted LRM outputs.

Table 10: Stability Study of Different judge Models.

| Model | GPT-4o-mini | o3-mini | Qwen-2.5-72B-instruction | Claude 3.7 Sonnet |
|---|---|---|---|---|
| IAA | 0.9 | 0.93 | 0.88 | 0.95 |
| Kappa | 0.79 | 0.81 | 0.74 | 0.85 |

Table 11: Step-Level Confusion Matrix: Claude 3.7 Sonnet vs. Human Judges

| Claude 3.7 Sonnet | Human Judges | |
|---|---|---|
| | Valid | Invalid |
| Valid | 782 | 20 |
| Invalid | 21 | 145 |

Table 12: Robustness of Judge Model and Metric to Noisy LRM Outputs.

| Model | Thought Num | | Efficiency | |
|---|---|---|---|---|
| | Original | Add noisy | Original | Add noisy |
| Claude-3.7-Sonnet | 0.28 | 0.27(-0.01) | 49.61% | 50.12%(+0.51%) |
| Deepseek-r1 | 9.17 | 9.17(0.0) | 48.96% | 49.93%(+0.93%) |
| Grok-3-mini-beta | 0.38 | 0.37(-0.01) | 61.69% | 62.64%(+0.95%) |

## F COST ANALYSIS

The experimental costs associated with the Think-Bench benchmark are primarily attributed to the judge model for evaluating the reasoning processes and CoT quality of the tested LRMs. When Claude 3.7 Sonnet was selected as the judge model, it was tasked with verifying key reasoning steps, assessing reflection quality, and computing efficiency metrics. For each LRM included in the evaluation, this judge model usage incurred an approximate cost of $40. In comparison, no additional costs are required if the open-source Qwen2.5-72B-instruction model is adopted as the judge model. It is important to note that this description excludes any expenses related to the tested

LRMs themselves (such as inference costs for generating model responses). The total cost incurred throughout the entire experimental process, encompassing data annotation, response generation by the tested LRMs, and the judge model-based evaluation, amounts to approximately $1,000.

## G    Limitations and future work

We believe our work still has room for improvement. First, although we have evaluated a broad set of state-of-the-art reasoning models, not all models natively support exposing their CoT content. As a result, some popular LRMs could not be assessed within our framework. Second, while our benchmark currently focuses on mathematics, physics, and chemistry, the framework can naturally be extended to other domains such as programming, law, or more open-ended reasoning tasks. Third, future work could incorporate interactive or multi-round reflection evaluation, providing a more holistic assessment of the role of reflection in reasoning efficiency and robustness.

## H    Use of LLMs

In this work, LLMs were employed as auxiliary tools in two ways. First, Claude 3.7 Sonnet was used to generate preliminary chain-of-thought reasoning chains as references and to judge the correctness and alignment of reasoning steps during evaluation (see Appendix I for details). Second, LLMs were used for minor language polishing of the manuscript to improve clarity and readability. Importantly, these models did not contribute to the conceptual design of the benchmark, the formulation of metrics, or the core scientific contributions. All research ideas, analyses, and substantive writing were conducted by the authors.

## I    Evaluation Prompts

---

**Key Steps Extraction Prompt**

You are an expert system that gives you a question and a corresponding answer, please list in detail the key reasoning steps from the question to the answer, make sure that the reasoning steps are clear and complete, and include all possible solutions. You should pretend not to know the basic truth answer beforehand.
Input: Question:
{question}
Answer:
{answer}
Output requirements:
1. Only include the essential key steps, and don't output unnecessary words
2. For each solution, record:
-logical_conclusion: The set of each key step of the solution, from Step 1 all the way to the answer
3. A problem may contain more than one way of reasoning, so make sure you don't miss any possible solutions.
4. Important: Output only JSON array with no additional information.
5. Don't add useless words to the process

Here is the json output format:
## Output format

```
[
{{
"solution1": {{
"logical_conclusion": ["step1:","step2:","step3:",...]
}}
}}
]
```

---

**Recall Evaluation Prompt**

# Task Overview
You are an expert system for verifying solutions to text-based problems. Your task is to match the ground truth middle steps with the provided solution.
# INPUT FORMAT:
1. Problem: The original question/task
2. A Solution of a model
3. Ground Truth: Essential steps required for a correct answer

# MATCHING PROCESS:
You need to evaluate each ground truth middle step against the solution, following these criteria:

## Match Criteria:
- **Exact Match or Equivalent Logical Step**: A ground truth step is considered **Matched** if:
- It appears exactly in the solution **OR**
- The same logical reasoning or idea is clearly expressed, even if wording or format differs.
- **Numerical and Conceptual Consistency**: All key numbers, equations, or transformations should align conceptually with the ground truth.
- **Step-by-Step Evaluation**: Each ground truth step must be assessed individually and sequentially.
- **Final Answer Check**: Judge whether the answer is correct.

# OUTPUT FORMAT:

```
[
  {{
    "step_index": <integer>,
    "judgment": "Matched" | "Unmatched",
    "correct_answer": "true" | "false"
  }}
]
```

# ADDITIONAL RULES:
1. **Strict JSON Output**: Output only the JSON array with no additional text or explanations.
2. **No Omitted Steps**: Every step in 'Ground Truth' must be evaluated.

# EDGE CASE HANDLING:
- If a step is conceptually equivalent but reworded, it is still considered **Matched**.
- If numerical transformations are equivalent (e.g., same formula in a different form), it is **Matched**.
- If the final answer is incorrect, `"correct_answer": "false"`.

Here is the problem, answer, solution, and the ground truth middle steps:

[Problem]
{question}
[Answer]
{answer}
[Solution]
{solution}
[Ground Truth Information]
{gt_annotation}

Precision Evaluation Prompt

# Task Overview
Given a solution with multiple reasoning steps for a text problem, reformat it into well-structured steps and evaluate their correctness.
# Step 1: Reformatting the Solution
Convert the unstructured solution into distinct reasoning steps while:
- Preserving all original content and order
- Not adding new interpretations
- Not omitting any steps

## Step Types
1. Logical Inference Steps
- Contains exactly one logical deduction
- Must produce a new derived conclusion
- Cannot be just a summary or observation

2. Background Information Steps
- External knowledge or question context
- No inference process involved

## Step Requirements
- Each step must be atomic (one conclusion per step)
- No content duplication across steps
- Initial analysis counts as background information
- Final answer determination counts as logical inference

# Step 2: Evaluating Correctness
Evaluate each step against:

## Ground Truth Matching
For logical inferences:
- Conclusion must EXACTLY match or be DIRECTLY entailed by ground truth

## Reasonableness Check (if no direct match)
Step must:
- Premises must not contradict any ground truth or correct answer
- Logic is valid
- Conclusion must not contradict any ground truth
- Conclusion must support or be neutral to correct answer

## Judgement Categories
- "Match": Aligns with ground truth
- "Reasonable": Valid but not in ground truth
- "Wrong": Invalid or contradictory
- "N/A": For background information steps

# Output Requirements
1. The output format MUST be in valid JSON format without ANY other content.
2. For highly repetitive patterns, output it as a single step.
3. Output maximum 35 steps. Always include the final step that contains the answer.
4. correct_answer: Whether the whole reasoning process produces the right answer.

Here is the json output format:
## Output Format

[

```
  {{
    "step_type": "logical inference|background information",
    "premise": "Evidence",
    "conclusion": "Step result",
    "judgment": "Match|Reasonable|Wrong|N/A"
    "correct_answer": "true|false"
  }}
]
```
Here is the problem, and the solution that needs to be reformatted to steps:

[Problem]
{question}
[Solution]
{solution}
[Correct Answer]
{answer}
[Ground Truth Information]
{gt_annotation}

---

### Model Output Reformat Prompt

I will present you with a solution to a problem. Unfortunately, the solution lacks proper paragraphing, making it hard to read. Your task is to improve readability by reformatting the solution into well-structured paragraphs. Follow these specific guidelines:
* Insert \n\n for paragraph breaks within the original solution. Do **NOT** alter any content of the original solution (the only exception is for itemized lists; see below).
- Each paragraph should represent a distinct, concise reasoning step that logically advances the solution.
- Reasoning steps can include case discussions, formula simplifications, or formula derivations. Each of these should be treated as an individual reasoning step and paragraphed accordingly.
- If an introductory analysis exists in the original solution, treat it as an initial reasoning step and place it as the first paragraph.
- Do **NOT** place any formulas in their own separate paragraphs; instead, include them within the same paragraph as the preceding text to form a cohesive reasoning step.

* For any itemized lists (ordered or unordered), convert them into a written format, such as "First/Second/Third." This is the **ONLY** content modification allowed.
* Avoid making paragraphs too lengthy, as long paragraphs might contain multiple reasoning steps that should be paragraphed separately.
* Disregard the accuracy of the solution content. Do **NOT** alter any of the original solution's content; focus solely on structuring it into logical, readable paragraphs.
* Reply with the reformatted solution directly.
———————————————————
Here is the problem, and the solution that needs to be reformatted:
[Problem]
{problem}
[Solution]
{response}

---

### First Correct Answer Extraction Prompt

The following is a problem and a solution (split into paragraphs, enclosed with tags and indexed from 0):
[Problem]

{problem}
[Correct Answer]
{answer}
[Solution]
{tagged_response}
Your task is to review and critique the solution paragraph by paragraph. Once you identify an correct answer in a paragraph, return the index of the paragraph where the earliest correct answer occurs. Otherwise, return the index of -1 (which typically denotes "not found"). Please put your final answer (i.e., the index) in
boxed.

## Reflection Quality Prompt

Here's a refined prompt that improves clarity and structure:
# Task Evaluate reflection steps in a problem-solving solutions, where reflections are self-corrections or reconsiderations of previous statements.
# Reflection Step Identification
Reflections typically begin with phrases like:
- "But xxx"
- "Alternatively, xxx"
- "Maybe I should"
- "Let me double-check"
- "Wait xxx"
- "Perhaps xxx"
It will throw an doubt of its previously reached conclusion or raise a new thought.
# Evaluation Criteria
Correct reflections must:
1. Reach accurate conclusions aligned with ground truth
2. Use new insights to find the mistake of the previous conclusion or verify its correctness.

Invalid reflections include:
1. Repetition - Restating previous content or method without new insights
2. Wrong Conclusion - Reaching incorrect conclusions vs ground truth
3. Incompleteness - Proposing but not executing new analysis methods
4. Other - Additional error types

# Input Format
[Problem]
{question}
[Think Content]
{think_content}
[Ground Truth]
{gt_annotation}
# Output Requirements
1. The output format must be in valid JSON format without any other content.
2. Output maximum 30 reflection steps.

Here is the json output format:
## Output Format

```
[
  {{
    "conclusion": "One-sentence summary of reflection outcome",
    "judgment": "Correct|Wrong",
    "error_type": "N/A|Repetition|Wrong Conclusion|
                   Incompleteness|Other"
  }}
```

```
]
# Rules 1. Preserve original content and order
2. No new interpretations
3. Include ALL reflection steps
4. Empty list if no reflections found
5. Direct JSON output without any other output
```

