# OpenReview forum: "THINK-Bench: Evaluating Thinking Efficiency and Chain-of-Thought Quality of Large Reasoning Models"
_ICLR.cc/2026/Conference — Submitted to ICLR 2026_

### Official Review · Reviewer_LYUk · 2025-10-30

**Soundness:** 2
**Presentation:** 3
**Contribution:** 2
**Rating:** 4
**Confidence:** 3

**Summary:**

This paper propose the Think-Bench benchmark, aiming to systematically evaluate the reasoning efficiency of CoT quality of large reasoning models. To accurately identify and quantitatively evaluate the thinking procedure of large reasoning models, the authors designed nine metrics to measure CoT efficiency and quality. Evaluation results on 11 LRMs show that most models exhibit overthinking behaviors.

**Strengths:**

S1: This work targets an important yet underexplored research question which is to evaluate the chain-of-thought (CoT) quality of large reasoning models. They have marched a step in quantitatively evaluating the efficiency and quality of CoTs.

S2: The collected dataset and designed metrics can be helpful for future research works to improve the CoT quality and efficiency.

**Weaknesses:**

W1: Limitations in model-generated and manually-annotated "groundtruth" intermediate steps. To evaluate the correctness of intermediate reasoning steps in LRM-generated solutions, the authors construct ground-truth key reasoning steps for all questions (Section 2.3). However, many scientific problems admit multiple valid solution paths—differing either in the sequence of reasoning steps or in the granularity with which the steps are decomposed. As a result, the proposed Recall and Precision metrics can only measure deviation from a single predefined “gold” reference, rather than alignment with the broader space of correct reasoning trajectories. The paper does not appear to account for this multi-solution nature, which raises concerns about the completeness and fairness of its evaluation framework.

W2: Several of the proposed evaluation metrics such as reflection quality, recall, and precision are based on judgments from Claude 3.7 Sonnet, itself a large reasoning model. However, the faithfulness of LRMs as evaluators has been questioned [1, 2]. Such models may hallucinate, produce inconsistent intermediate judgments, or even contradict their own final conclusions. The paper should include a more thorough discussion of this reliability issue and clarify whether additional validation or calibration steps were taken to ensure the trustworthiness of the LRM-based judgments.

---
Reference

[1] Chen Y, Benton J, Radhakrishnan A, et al. Reasoning Models Don't Always Say What They Think[J]. arXiv preprint arXiv:2505.05410, 2025.

[2] Lu J, Xu Z, Kankanhalli M. Reasoning LLMs are Wandering Solution Explorers[J]. arXiv preprint arXiv:2505.20296, 2025.

**Questions:**

Q1. The valid reflection seems to have a vague and arguable definition (Line 256 - Line 258). Can you give an example what is a valid reflection which confirm an earlier conclusion? What is an invalid reflection by contrast?

Q2. Several evaluation metrics are quite "subjective" to specific judges - if changing the judge to different models, are the evaluation results still consistent?

---

> ### Author Response · Authors · 2025-11-21
> **Response to W1, W2, Q1 and Q2:**
>
> **Response to W1:**
>
> We sincerely appreciate you raising this important concern regarding the fairness of the evaluation framework with respect to multiple problem-solving paths. We acknowledge that many scientific problems can be solved through multiple correct approaches. Therefore, during the dataset construction phase, we instructed annotators to identify and label 2 to 4 of the most common and representative problem-solving paths. This represents a balanced strategy between ensuring annotation quality and covering prevalent solution methods.
>
> More importantly, current large language models (LLMs) are trained based on human knowledge and experience, and their reasoning processes are designed to align with the thinking patterns of human experts. In our experiments, we did not impose any specific guidance on the reasoning processes of LLMs. Consequently, we observed a high degree of consistency between the naturally generated problem-solving paths of LLMs and the set of annotated common paths.
>
> For further verification, we conducted random sampling validation on a subset of questions and models with low Recall and Precision scores. The results indicate that the logical frameworks of the model-generated paths are consistent with our annotated paths. The primary reasons for lower scores are that the models failed to fully execute certain key steps or omitted necessary steps, rather than adopting entirely different, unannotated problem-solving paths. Due to the length of the model outputs, we cannot directly provide specific model responses as examples in this reply. However, all raw model outputs have been uploaded to an anonymous link(https://anonymous.4open.science/r/Think-Bench-anony-6866), which is accessible for review at any time.
>
> **Response to W2 and Q2:**
>
> Thank you for your thoughtful observation regarding the reliance on Claude.
> We would like to correct a potential misinterpretation: Claude 3.7 Sonnet in our framework only performs step alignment (matching model-generated reasoning steps to human-annotated key steps), not direct judgment of step correctness. The correctness of steps is pre-defined by human-annotated ground truth, and Claude’s role is to check if model steps semantically align with this ground truth. To address concerns about Claude’s reliability, we conducted a dedicated stability test (detailed in Appendix E.2 and E.3, lines 1118-1176): We compared Claude’s step-alignment results with human judgments on 100 instances (968 steps), achieving 95% agreement and a Cohen’s Kappa of 0.85. This confirms strong consistency between Claude’s evaluations and human expertise, even for relatively complex reasoning steps. Additionally, we have conducted relevant experiments on the selection of judge models, which are presented in the Appendix. As shown in the following table, Claude 3.7 Sonnet achieved results more consistent with human judgments, and thus we selected it as the judge model for our experiments.
> |Model|GPT-4o-mini|o3-mini|Qwen-2.5-72B-instruction|Claude3.7Sonnet|
> |-----|-----------|-------|------------------------|---------------|
> |IAA|0.9|0.93|0.88|0.95|
> |Kappa|0.79|0.81|0.74|0.85|
>
> |Claude3.7Sonnet|HumanJudges-Valid|HumanJudges-Invalid|
> |---------------|-----------------|-------------------|
> |Valid|782|20|
> |Invalid|21|145|
>
> **Response to Q1:**
>
> We agree that clarifying this concept with explicit criteria and examples is critical to enhancing the rigor of our work.
> We would like to further elaborate on the definition of valid and invalid reflection, aligning with the evaluation framework detailed in the text: A valid reflection is defined by two core scenarios:  (1) it explicitly identifies and corrects errors in prior reasoning by introducing new analytical perspectives (e.g., re-examining a miscalculated formula with alternative mathematical transformations) or supplementary evidence (e.g., citing additional principles to verify a derivation);  (2) it draws a confirmatory conclusion consistent with the annotated ground truth through additional reasoning (e.g., after initially deriving a result via one method, re-verifying the same conclusion using a distinct, logically sound approach to reinforce reliability).  In contrast, a invalid reflection includes cases such as: (1) mere repetition of existing content (e.g., restating "the answer is 29 cents" without any new verification steps after already concluding this);  (2) deriving conclusions inconsistent with the ground truth (e.g., incorrectly revising a correct intermediate result due to flawed logic);  (3) proposing but failing to execute new analytical methods (e.g., suggesting "we could check via unit conversion" but not following through with the calculation);  or (4) other errors that do not contribute to valid reasoning.
> To make this distinction more concrete, we have supplemented explicit examples of valid and invalid reflection in Appendix (corresponding to Figure 9 in the revised manuscript).

---

> ### Author Response · Authors · 2025-11-27
>
> Dear Reviewer LYUk,
>
> Thank you very much for the time and effort you dedicated to reviewing our work and for the valuable comments you provided. We carefully considered all your feedback, addressed the concerns in our response, and revised our paper accordingly.
>
> We sincerely hope that our clarifications and additional experiments help resolve your concerns. If you have any further concerns or questions, please let us know and we are more than happy to further address them.
>
> Thank you again for your thoughtful and constructive review.
>
> Best regards,
>
> The Authors

---

### Official Review · Reviewer_gcSF · 2025-10-30

**Soundness:** 3
**Presentation:** 3
**Contribution:** 3
**Rating:** 6
**Confidence:** 3

**Summary:**

This paper proposes Think-Bench, a benchmark specifically designed to evaluate the thinking efficiency and CoT quality of LRMs. The dataset contains 1,375 questions covering mathematics, physics, and chemistry, each with manually annotated key inference steps, totaling over 13,000 annotations. The authors designed a comprehensive evaluation metric system, including six efficiency metrics such as total tokens, first-time correct tokens, efficiency, reflection tokens, and number of thinks, and two CoT quality metrics. Through evaluation of 11 representative LRMs, the paper reveals the prevalent over thinking problem in current models, particularly in handling simple problems, where excessively long inference chains are generated, leading to wasted computational resources.

**Strengths:**

S1. The problem addressed in this paper is indeed a real challenge faced by large-scale inference models, especially given the increasing popularity of test-time-scaling strategies. This paper not only focuses on the accuracy of the final answer but also delves into the efficiency and quality of the reasoning process, filling a significant gap in existing evaluation systems. The proposed six efficiency indicators and two quality indicators complement each other, forming a relatively complete evaluation system capable of characterizing the model's reasoning behavior from multiple perspectives.

S2. The paper employs a rigorous two-stage annotation process. First preliminary references are generated using Claude 3.7 Sonnet, and then a team of graduate students with backgrounds in mathematics, physics, and chemistry conducts manual review and corrections. This annotation method ensures both efficiency and quality.

S3. The paper evaluates 11 representative LRMs, covering models of different sizes from 1.5B to 235B parameters and types, with broad experimental coverage. The experiments not only report overall performance but also provide detailed analyses by difficulty level and subject, revealing some interesting patterns.

S4. The paper provides an anonymous code repository containing the complete dataset, evaluation code, and all experimental scripts, which is good for the community.

**Weaknesses:**

W1. The core assessment in this paper relies on Claude 3.7 Sonnet to determine the correctness of reasoning steps and their match with the reference answer, but the reliability of this assessor is not adequately verified. There is no report on the inter-annotator agreement between human assessment and LLM assessment, nor is there an analysis of potential systematic biases introduced by Claude. For example, Claude may have a preference for certain expression styles or reasoning paths, which could affect the fairness of the assessment.

W2. What kind of reasoning chain is considered "excessive"? Is it merely a matter of length, or does it also include repetitiveness and inefficiency? The paper primarily uses the percentage of correctly identified tokens on the first attempt to measure this, but this metric has significant limitations: if the model performs necessary exploration before providing the correct answer on the first attempt, is this also considered "excessive"?

W3. The criteria for judging "reflective quality" are rather subjective and lack a clear operational definition. These conceptual ambiguities may lead to inconsistent understandings of the results among different researchers.

W4. ThinkBench only covers three subjects: mathematics, physics, and chemistry, which limits the representativeness of the benchmark. Many important reasoning scenarios, such as programming, legal reasoning, and common sense reasoning, are not covered. Is it because annotating key steps is not feasible for issues that are not fully well-defined?

W5. Section 4.3 mentions that some models produce empty outputs. The paper chooses to skip these samples directly to "maintain the integrity of the evaluation," but this approach may actually introduce bias. These failure cases themselves reflect the model's deficiencies in reasoning ability, and simply skipping them will make the evaluation results of these models appear better than they actually are.

W6. The paper mentions some possible directions for improvement in the conclusion such as dynamic inference paths and early exit mechanisms, but these are all general and lack specific technical solutions or preliminary experimental verification. Including ablation experiments or case studies to explore which factors affect inference efficiency would make the paper more in-depth and instructive.

W7. The paper's evaluation process relies on multiple calls to Claude 3.7 Sonnet for step matching and quality assessment, which is quite expensive in terms of both computation and time. This could be a practical obstacle for researchers who want to evaluate new models using Think-Bench. The paper does not report the total cost including API call fees, time consumption and so forth.

**Questions:**

Q1. Did you conduct manual verification to check the accuracy of Claude 3.7 Sonnet as a judge?

Q2. Regarding the empty output issue mentioned in Section 4.3, could you provide more detailed statistics: How frequently does each model produce empty outputs? How does the distribution of this problem differ between simple and hard problems? Is it related to a specific subject or problem type? Why are these cases not counted as errors but instead skipped? Does this approach overestimate the actual performance of these models? How would the rankings change if empty outputs were recalculated and counted as errors?

Q3. The efficiency metric is defined as the percentage of the total length where the first correct answer appears, but does this truly reflect the "efficiency" of the reasoning? Consider two scenarios: First, the model quickly provides the correct answer and then performs extensive verification; Second, the model provides the correct answer only after careful consideration. By your definition, scenario 1 is more efficient, but the verification process in scenario 2 may be necessary, especially for difficult problems. How do you view this trade-off?

Q4. The `reflection quality` metric requires judging whether a reflection step accurately identified previous errors or provided new insights, and these criteria are quite subjective. Different people may have different judgments. Have you conducted inter-annotator consistency checks? Can you provide some boundary cases illustrating what constitutes effective reflection and what constitutes ineffective reflection? Is this metric reliable enough?

---

> ### Author Response · Authors · 2025-11-21
> **Response to W1, W2, W3, W4, Q1, Q3 and Q4:**
>
> **Response to W1 and Q1:**
>
> Thank you for your thoughtful observation regarding the reliance on Claude.
> We would like to correct a potential misinterpretation: Claude 3.7 Sonnet in our framework only performs step alignment (matching model-generated reasoning steps to human-annotated key steps), not direct judgment of step correctness. The correctness of steps is pre-defined by human-annotated ground truth, and Claude’s role is to check if model steps semantically align with this ground truth. To address concerns about Claude’s reliability, we conducted a dedicated stability test (detailed in Appendix E.2 and E.3, lines 1118-1176): We compared Claude’s step-alignment results with human judgments on 100 instances (968 steps), achieving 95% agreement and a Cohen’s Kappa of 0.85. This confirms strong consistency between Claude’s evaluations and human expertise, even for relatively complex reasoning steps. Additionally, we have conducted relevant experiments on the selection of judge models, which are presented in the Appendix. As shown in the following table, Claude 3.7 Sonnet achieved results more consistent with human judgments, and thus we selected it as the judge model for our experiments.
>
> |Model|GPT-4o-mini|o3-mini|Qwen-2.5-72B-instruction|Claude3.7Sonnet|
> |-----|-----------|-------|------------------------|---------------|
> |IAA|0.9|0.93|0.88|0.95|
> |Kappa|0.79|0.81|0.74|0.85|
>
> |Claude3.7Sonnet|HumanJudges-Valid|HumanJudges-Invalid|
> |---------------|-----------------|-------------------|
> |Valid|782|20|
> |Invalid|21|145|
>
> **Response to W2 and Q3:**
>
> We appreciate your detailed scrutiny of our metrics.  Regarding the definition of "excessive" and the efficiency trade-off, we empirically examined the inference patterns of all evaluated LRMs and observed that they strictly adhere to a "Reasoning-then-Answer" paradigm, rather than the "Answer-then-Verify" pattern (Scenario 1).  Consequently, the concern that our metric penalizes "quick answers followed by necessary verification" does not apply to the actual behavior of current reasoning models.  Based on this, our metric, defined as $Efficiency = \text{First Correct Tokens} / \text{Total Tokens}$ (lines 244-247), is robust: (1) Necessary Exploration is Rewarded: If a model engages in extensive exploration before answering, both the numerator (First Correct Tokens) and denominator increase proportionally, maintaining a high efficiency score (approaching 1.0);  (2) Overthinking is Penalized: The metric specifically targets redundancy generated after the correct answer is reached (where the numerator stops growing but the denominator increases), which aligns with our definition of overthinking as "excessive and redundant tokens" rather than valid exploratory reasoning.
>
> **Response to W3 and Q4:**
>
> We agree that clarifying this concept with explicit criteria and examples is critical to enhancing the rigor of our work.
> We would like to further elaborate on the definition of valid and invalid reflection, aligning with the evaluation framework detailed in the text: A valid reflection is defined by two core scenarios:  (1) it explicitly identifies and corrects errors in prior reasoning by introducing new analytical perspectives (e.g., re-examining a miscalculated formula with alternative mathematical transformations) or supplementary evidence (e.g., citing additional principles to verify a derivation);  (2) it draws a confirmatory conclusion consistent with the annotated ground truth through additional reasoning (e.g., after initially deriving a result via one method, re-verifying the same conclusion using a distinct, logically sound approach to reinforce reliability).  In contrast, a invalid reflection includes cases such as: (1) mere repetition of existing content (e.g., restating "the answer is 29 cents" without any new verification steps after already concluding this);  (2) deriving conclusions inconsistent with the ground truth (e.g., incorrectly revising a correct intermediate result due to flawed logic);  (3) proposing but failing to execute new analytical methods (e.g., suggesting "we could check via unit conversion" but not following through with the calculation);  or (4) other errors that do not contribute to valid reasoning.
> To make this distinction more concrete, we have supplemented explicit examples of valid and invalid reflection in Appendix (corresponding to Figure 9 in the revised manuscript).
>
> **Response to W4:**
>
> Thank you for your thoughtful observation regarding the domain coverage, we focused on three disciplines to ensure high-quality, human-verified annotations.   We agree that broader coverage is valuable;   however, our annotation resources were limited.  It is inherently impossible for a single benchmark to encompass the full spectrum of all possible reasoning domains.  The benchmark is designed to be easily extensible, and we welcome community-driven or follow-up efforts to expand it to additional domains.

---

> ### Author Response · Authors · 2025-11-21
> **Response to W5 , W6, W7 and Q2:**
>
> **Response to W5 and Q2:**
>
> We sincerely apologize for the ambiguity in our original phrasing regarding "maintaining the integrity of the evaluation," which led to the misunderstanding that we discarded empty outputs. In reality, we strictly treated all empty responses as incorrect answers in our evaluation rather than skipping them, and we have updated the manuscript to explicitly reflect this handling method (lines 459-461);therefore, the reported performance and rankings do not overestimate the models' capabilities. Regarding the specific statistics, the frequency of empty outputs is extremely low (less than 0.5%) and was observed exclusively in the DeepSeek distilled series and Ernie-x1-turbo-32k. Our analysis indicates that these failures appear sporadically without any specific correlation to subject matter or problem difficulty. We have also communicated with the respective model providers, who confirmed that these are likely sporadic internal or deployment-side anomalies currently under investigation, and given their negligible frequency combined with our strict penalization strategy, they do not introduce bias into the overall benchmarking results.
>
> **Response to W6:**
>
> Thank you for your feedback. We would like to clarify that the directions mentioned in the conclusion, such as dynamic inference paths and early exit mechanisms, are presented as promising suggestions for future research derived from our empirical findings, rather than the core research content of this specific paper. As a benchmark study, our primary objective is to rigorously identify and quantify the overthinking phenomenon through Think-Bench. These discussion points serve as a roadmap to inspire the community, providing a clear direction for other researchers to utilize our benchmark to develop and validate specific mitigation strategies in follow-up work.
>
> **Response to W7:**
>
> Thank you for your valuable feedback concerning the practical cost of using Think-Bench. We have updated Appendix F to explicitly report the detailed expenses, showing that the cost per evaluated model using Claude 3.7 Sonnet as the judge is under 40 dollar, which we believe is reasonable for a academic benchmarking. More critically, the judge model is indeed replaceable; open-source alternatives with strong reasoning capabilities can be adopted at no additional cost. However, we must note a critical trade-off: our selection of Claude 3.7 Sonnet was based on its demonstrated superior alignment with human annotation, ensuring the highest fidelity in scoring the complex key steps of reasoning chains. See Appendix E for detailed experimental details. Details of the experiment on costs are as follows (lines 1180-1190):
> The experimental costs associated with the Think-Bench benchmark are primarily attributed to the judge model for evaluating the reasoning processes and CoT quality of the tested LRMs. When Claude 3.7 Sonnet was selected as the judge model, it was tasked with verifying key reasoning steps, assessing reflection quality, and computing efficiency metrics. For each LRM included in the evaluation, this judge model usage incurred an approximate cost of 40 dollar. In comparison, no additional costs are required if the open-source Qwen2.5-72B-instruction model is adopted as the judge model. It is important to note that this description excludes any expenses related to the tested LRMs themselves (such as inference costs for generating model responses). The total cost incurred throughout the entire experimental process, encompassing data annotation, response generation by the tested LRMs, and the judge model-based evaluation, amounts to approximately 1,000 dollar.

---

> ### Author Response · Authors · 2025-11-27
>
> Dear Reviewer gcSF,
>
> Thank you very much for the time and effort you dedicated to reviewing our work and for the valuable comments you provided. We carefully considered all your feedback, addressed the concerns in our response, and revised our paper accordingly.
>
> We sincerely hope that our clarifications and additional experiments help resolve your concerns. If you have any further concerns or questions, please let us know and we are more than happy to further address them.
>
> Thank you again for your thoughtful and constructive review.
>
> Best regards,
>
> The Authors

---

### Official Review · Reviewer_fUgn · 2025-10-31

**Soundness:** 2
**Presentation:** 3
**Contribution:** 2
**Rating:** 4
**Confidence:** 4

**Summary:**

The paper introduces Think-Bench, a benchmark to evaluate reasoning efficiency and the quality of CoT in LLMs. It aims to evaluate the overthinking problem. The paper introduces new efficiency metrics and conducts an empirical evaluation, highlighting inefficiencies in models despite high CoT quality.

**Strengths:**

1. The idea of evaluating LRMs on both reasoning efficiency and CoT quality is innovative, providing a new framework that looks beyond the final answer accuracy, which is a limitation of many existing benchmarks.

2. The paper performs a detailed evaluation of 11 models, including both proprietary and open-source LRMs, across various domains and difficulty levels, offering insights into current reasoning inefficiencies.

3. The proposed metrics for reasoning quality, like reflection quality, contribute valuable tools to the field, facilitating further exploration of reasoning models.

**Weaknesses:**

1. Numerous studies have already proposed solutions to address overthinking [1-3], such as dynamic reasoning path design or early-exit mechanisms. This paper merely cites these ideas as potential future directions without providing any concrete comparative analysis, resulting in a rather weak analytical foundation.

2. The definition of “efficiency” is quite ambiguous, and it is unclear which type of efficiency the paper refers to. For instance, if efficiency is defined from the user’s perspective, it should correspond to the number of tokens required to produce a correct answer. If it concerns the model’s own efficiency, it should logically be defined as the number of tokens required to generate the correct answer on the first attempt (i.e., the number of tokens spent in a successful initial exploration, which aligns more closely with the conventional notion of efficiency—the smaller, the better). Dividing this by the total sequence length seems conceptually inconsistent. Moreover, if the model arrives at the correct answer in the first attempt but requires three to four rounds of verification, this could still indicate efficient reasoning and exploration, with inefficiency lying instead in the critic’s quality or performance.

3. The method heavily depends on Claude’s critic capability. When facing highly complex problems that exceed the model’s reasoning boundaries, even if intermediate reasoning steps contain errors, the model may fail to detect them. A rigorous evaluation framework should take this issue into account.

4. The evaluation cost is extremely high, and since the approach relies on closed-source models, reproducibility becomes difficult. The robustness of the results also remains uncertain.

5. Most of the performance analysis is descriptive, lacking an in-depth examination of the trade-off between reasoning quality and efficiency. Although the paper claims to demonstrate a balance, it does not sufficiently discuss why certain models excel in efficiency while others perform better in reasoning quality—particularly in tasks such as GSM8K, where direct instruction-following models may actually be more effective.

6. The error analysis in Section 4.3 focuses only on specific failure cases in a few models. A more comprehensive exploration of error types (e.g., logical inconsistencies or misinterpretations of the problem) would yield deeper insights into the limitations of these models.

7. In fact, all traditional CoT benchmarks seem to be able to perform similar metrics for evaluation by Claude, and the data contribution in this work does not appear to be that important.

[1] Sui Y, Chuang Y N, Wang G, et al. Stop overthinking: A survey on efficient reasoning for large language models[J]. arXiv preprint arXiv:2503.16419, 2025.

[2] Shen Y, Zhang J, Huang J, et al. Dast: Difficulty-adaptive slow-thinking for large reasoning models[J]. arXiv preprint arXiv:2503.04472, 2025.

[3] Chen Q, Peng D, Liu J, et al. Aware First, Think Less: Dynamic Boundary Self-Awareness Drives Extreme Reasoning Efficiency in Large Language Models[J]. arXiv preprint arXiv:2508.11582, 2025.

**Questions:**

1. This paper should mention more concrete comparative analysis about efficient reasoning.

2. What exactly does the term refer to efficiency from the user’s perspective, or the model’s intrinsic efficiency? Can you further explain why you define the efficiency in such way.

3. Since the proposed method heavily depends on Claude’s critic capability, how does the evaluation framework handle cases where the task complexity exceeds the model’s reasoning boundary? If the critic fails to detect errors in intermediate steps, how can the reliability of the evaluation be ensured?

4. The paper should discuss how to mitigate these issues, or assess the robustness of its results under different evaluation settings.

5. Can you provide deeper analysis on why there is no deeper examination of the trade-off between reasoning quality and efficiency?

6. In Section 4.3, the error analysis focuses only on specific failure cases. Why does the paper not include a broader categorization of error types, which could offer more comprehensive insights into the models’ limitations?

---

> ### Author Response · Authors · 2025-11-21
> **Response to W1, W2, W3, W4, W7, Q2 and Q3**
>
> **Response to W1:**
>
> Thank you for pointing out these related works. We agree that comparisons would be valuable. However, the three referenced papers cannot be evaluated in a reproducible way.    The first is a survey and does not provide a concrete model or standardized implementation, identifying a comparable system from it is extremely difficult. The second offers code but no model checkpoints, and we cannot retrain it within our time and resource limits.  The third provides only an anonymous and incomplete code release that cannot run. Because none of these works provide usable or reproducible models, controlled comparison is currently infeasible.
>
> **Response to W2 and Q2:**
>
> Thank you for your valuable feedback on the Efficiency definition. First, we confirm the Efficiency in our work refers explicitly to the model’s intrinsic reasoning efficiency , consistent with your understanding of "tokens for first correct answer." As defined in Equation (1) (lines 244-248): Efficiency = $\hat{T}_i/T_i$, where $\hat{T}_i$ is tokens to the first correct answer, and $T_i$ is total reasoning tokens ($\hat{T}_i=0$ if no correct answer).
> The ratio quantifies how much of the model’s total token consumption contributes to achieving early correct convergence (vs. redundant post-verification). For example, two models with $\hat{T}_i=500$ differ if $T_i=600$ (Efficiency=83.3%) vs. $T_i=1200$ (Efficiency=41.7%).
>
> **Response to W3 and Q3:**
>
> Thank you for your thoughtful observation regarding the reliance on Claude.
> We would like to correct a potential misinterpretation: Claude 3.7 Sonnet in our framework only performs step alignment (matching model-generated reasoning steps to human-annotated key steps), not direct judgment of step correctness. The correctness of steps is pre-defined by human-annotated ground truth, and Claude’s role is to check if model steps semantically align with this ground truth. To address concerns about Claude’s reliability, we conducted a dedicated stability test (detailed in Appendix E.2 and E.3, lines 1118-1176): We compared Claude’s step-alignment results with human judgments on 100 instances (968 steps), achieving 95% agreement and a Cohen’s Kappa of 0.85. This confirms strong consistency between Claude’s evaluations and human expertise, even for relatively complex reasoning steps. Additionally, we have conducted relevant experiments on the selection of judge models, which are presented in the Appendix. As shown in the following table, Claude 3.7 Sonnet achieved results more consistent with human judgments, and thus we selected it as the judge model for our experiments.
> |Model|GPT-4o-mini|o3-mini|Qwen-2.5-72B-instruction|Claude3.7Sonnet|
> |-----|-----------|-------|------------------------|---------------|
> |IAA|0.9|0.93|0.88|0.95|
> |Kappa|0.79|0.81|0.74|0.85|
>
> |Claude3.7Sonnet|HumanJudges-Valid|HumanJudges-Invalid|
> |---------------|-----------------|-------------------|
> |Valid|782|20|
> |Invalid|21|145|
>
> **Response to W4:**
>
> Thank you for your valuable feedback on cost and reproducibility. First, regarding evaluation cost: Our updated Appendix F explicitly reports detailed expenses, showing the cost per evaluated model is under $40 when using Claude 3.7 Sonnet as the judge. More importantly, the judge model is replaceable: Open-source alternatives with strong reasoning capabilities can be adopted at no additional cost. On reproducibility: We have made the full pipeline (dataset, evaluation scripts) publicly available via the anonymous repository (https://anonymous.4open.science/r/Think-Bench-anony-6866), enabling researchers to replicate results with either closed-source or open-source judges. Regarding robustness, our work focuses on evaluating the efficiency and chain-of-thought quality of LRMs.  Robustness is indeed an important research direction, but it falls outside the scope of this paper. Our goal in this work is to provide a solid foundation for efficiency and CoT-quality assessment, and we hope future work can build on this framework to explore robustness more fully.
>
> **Response to W7:**
>
> Thank you for your feedback regarding our contribution to the dataset for our project.  A key distinction between our Think-Bench and traditional benchmarks is the inclusion of fine-grained, step-level annotations, which existing benchmarks (such as MMLU and GPQA) do not provide. These annotations help support our proposed metrics by aligning the steps generated by the model with essential logical components that have been verified by humans. This level of granularity is not achievable with benchmarks that only annotate final answers. Thus, our work contributes both novel evaluation metrics (such as Efficiency, Recall, Precision, Reflection Quality, etc.) and a dataset with step-level annotations, addressing a gap where traditional CoT benchmarks fall short in supporting rigorous, process-oriented evaluations of reasoning efficiency and CoT quality.

---

> ### Author Response · Authors · 2025-11-21
> **Response to W5, Q1 and Q5**
>
> **Response to W5, Q1 and Q5:**
>
> Thank you for your feedback on strengthening the quality-efficiency trade-off analysis and comparative discussion.  We have included in-depth analysis of the quality-efficiency trade-off in Appendix C of the revised manuscript.  This section explicitly explains why models differ in balancing efficiency and reasoning quality. Details are as follows (lines 810-910):
>
> **In-depth Analysis Of The Results**
>
> We conduct an in-depth analysis of the experimental results by integrating the specific mechanisms of model training and decoding design. First, Grok-3-mini-beta, one of the most efficient models in the experiment, exhibits a small number of Thought Num, low Reflection Tokens, and the highest Efficiency. The core reason for this phenomenon lies in the fact that the Grok-3 series models extensively adopt reinforcement learning or targeted strategy optimization during training to refine reasoning paths, enabling the model to acquire the ability to achieve goals or complete necessary self-verification through fewer steps. This further reduces redundant generation and directly improves the ratio of First-Correct Tokens to Total-Tokens.
>
> In contrast, models with better performance in CoT quality metric, such as Qwen3-235b-a22b and Ernie-x1-turbo-32k, tend to adopt or allow prolonged extended thinking or multi-path sampling strategies. These models generate multiple or longer reasoning chains through test-time scaling, self-consistency sampling, or explicit reflection or verification mechanisms, and integrate answers via majority voting to enhance the stability and accuracy of results. Relevant existing studies have confirmed that self-consistency sampling can significantly improve the correctness of final answers, but it comes at the cost of generating more tokens, thereby reducing Efficiency.
>
> Furthermore, distilled or pruned variant models demonstrate low Recall and Precision in the experiment, which is consistent with the inherent characteristic of the distillation process. Although distillation technology helps improve model throughput and the explicit conciseness of outputs, it is prone to losing the completeness of intermediate reasoning steps or failing to fully retain alignable key reasoning components, ultimately leading to a decline in CoT quality. The experimental data table also shows that the efficiency of small distilled models is not always superior, while their quality decreases significantly. Finally, some models extend reasoning chains when necessary through extended thinking modes or controllable thinking budget mechanisms. This design explains why Claude can maintain high reflection quality while using a relatively small number of total tokens: the model implements a scheduling mechanism of slowing down for verification when necessary and terminating reasoning as early as possible when unnecessary, thus achieving a balanced advantage between efficiency and quality.
>
> Figure 7 intuitively illustrates a prominent token complexity phenomenon~\citep{lee2025well}. First, with the increase in problem difficulty and the model’s tendency toward in-depth verification, the total number of tokens exhibits a non-linear growth. Specifically, a large number of additional reflection or verification steps do not necessarily lead to a proportional improvement in accuracy. This trend is observed across multiple large language models, with accuracy plateauing in the high token consumption range. Second, when analyzing with First Correct Tokens as the horizontal axis, the curves of different models demonstrate the critical role of early correct convergence in reasoning efficiency. When the First Correct Tokens are low and the Efficiency is high, the model can significantly reduce the total computational cost while maintaining accuracy. In contrast, scenarios with both high First Correct Tokens and high Reflection Tokens often correspond to high accuracy but extremely high token complexity, a situation that is more prone to inducing overthinking especially for simple problems.

---

> ### Author Response · Authors · 2025-11-21
> **Response to W6, Q4 and Q6**
>
> **Response to W6 and Q6:**
>
> Thank you for the insightful suggestion. In the original submission, Section 4.3 concentrated on technical failure cases, such as certain DeepSeek distilled models generating only intermediate reasoning without final answers. These cases were sporadic, not tied to intrinsic reasoning limitations, and were confirmed by the model teams as deployment-related issues under investigation.
> Following your suggestion, we have expanded the error analysis to include a broader examination of actual reasoning errors made by LRMs (lines 446-452). In the revised manuscript and the appendix, we provide representative examples covering common reasoning failures, including misinterpretation of key problem conditions, incorrect application of formulas or theorems, and flawed logical derivations. As shown in the new example in Appendix D (Figure 11), a typical pattern involves misunderstanding multi-scenario constraints and failing to consider all valid boundaries, which leads to one-sided reasoning and incorrect final conclusions.
>
> **Response to Q4:**
>
> Thank you for your thoughtful observation, to address your suggestion regarding mitigation strategies, we conducted an additional experiment utilizing a prompt engineering technique with the specific instruction "Answer this question with minimal thought:" to assess its impact on reasoning overhead. The results demonstrate that this simple constraint effectively mitigates overthinking behaviors by significantly reducing token consumption; for example, Deepseek-r1 and Qwq-plus exhibited token reductions of 24.41% and 20.46%, respectively. Furthermore, this approach led to observable improvements in our Efficiency metric for several models, such as a 2.3% increase for Ernie-x1-turbo-32k and a 2.28% increase for Qwq-plus, confirming that Think-Bench can effectively quantify the positive impact of such mitigation methods on reasoning process efficiency. Regarding robustness, our work focuses on evaluating the efficiency and chain-of-thought quality of LRMs.  Robustness is indeed an important research direction, but it falls outside the scope of this paper. Our goal in this work is to provide a solid foundation for efficiency and CoT-quality assessment, and we hope future work can build on this framework to explore robustness more fully.
> |Modelname|Efficiency|Efficiency-PromptEngineering|Tokens|Tokens-PromptEngineering|
> |---------|-------------|-----------------------------|---------|------------------------|
> |Claude-3-7-sonnet|49.61\%|49.17\%(-0.44\%)|942.82|862.78(-8.49\%)|
> |Deepseek-r1|48.96\%|50.36\%(+1.4\%)|3795.19|2868.89(-24.41\%)|
> |Grok-3-mini-beta|61.69\%|63.27\%(+1.58\%)|1891.34|1665.52(-11.94\%)|
> |Ernie-x1-turbo-32k|47.02\%|49.32\%(+2.3\%)|4692.21|4150.88(-11.54\%)|
> |Qwq-plus|44.58\%|46.86\%(+2.28\%)|5738.37|4564.18(-20.46\%)|
> |Glm-z1-air|47.41\%|46.58\%(-0.83\%)|3678.68|2986.13(-18.83\%)|

---

> ### Author Response · Authors · 2025-11-27
>
> Dear Reviewer fUgn,
>
> Thank you very much for the time and effort you dedicated to reviewing our work and for the valuable comments you provided. We carefully considered all your feedback, addressed the concerns in our response, and revised our paper accordingly.
>
> We sincerely hope that our clarifications and additional experiments help resolve your concerns. If you have any further concerns or questions, please let us know and we are more than happy to further address them.
>
> Thank you again for your thoughtful and constructive review.
>
> Best regards,
>
> The Authors

---

### Official Review · Reviewer_oQGU · 2025-11-02

**Soundness:** 3
**Presentation:** 4
**Contribution:** 2
**Rating:** 4
**Confidence:** 3

**Summary:**

The paper introduces Think-Bench, a benchmark for evaluating reasoning efficiency and chain-of-thought (CoT) quality in LRMs. It contributes (i) a dataset with human-curated key steps across math/physics/chemistry, (ii) a suite of efficiency metrics (e.g., First-Correct Tokens, Reflection Tokens, Thought Num) plus CoT recall/precision, and (iii) a multi-model empirical study showing widespread “overthinking,” especially on easy questions.

**Strengths:**

Process-level evaluation. Goes beyond final accuracy to step-level CoT quality (recall/precision), enabled by human key-step annotations and a judging pipeline. This addresses a real gap in current benchmarks.

Clear efficiency metrics. The definitions of Efficiency (first-correct/total), Reflection Tokens, and Thought Num are intuitive and operationalizable.

Empirical findings. Consistent evidence of “overthinking,” differences across subjects and difficulty, and model-specific trade-offs are well documented (Tables 2–6, Fig. 6).

**Weaknesses:**

1. Missing marquee closed models. The model list includes Claude 3.7 Sonnet, DeepSeek-R1(+distills), Qwen3-235B, Grok-3-mini, ERNIE-X1-Turbo, GLM-Z1-Air—but not GPT-4/5 or Gemini-2.5-Pro. For a paper about thinking efficiency of LRMs, omission of today’s most-used LLMs weakens external validity. Authors do note some models don’t expose CoT, but a discussion/ablation on “no-CoT models” or proxy evaluations would help.

2. Simplicity / metric specificity. While practical, some metrics (e.g., Thought Num via discourse markers) feel heuristic and dataset-specific. The work would be stronger if it tied metrics to theoretical desiderata (e.g., optimal stopping/early-exit) or validated metric stability under paraphrase/noise.

3. Adoption risk. The benchmark is limited to STEM QA (1,375 items) and relies on Claude-as-judge for step matching. Community uptake may hinge on (a) openness of the judging pipeline beyond a single vendor, and (b) extensibility beyond the three domains. The paper mentions extensibility but does not demonstrate it.

4. Human reasoning annotations enable interpretable, step-aware evaluation rather than outcome-only accuracy. However, inter-annotator agreement and robustness of the step set (multiple valid paths) need more quantification; currently it’s described but not stress-tested.

**Questions:**

- Can you justify model coverage (why omit GPT-4/5, Gemini-2.5-Pro)? If CoT visibility is the blocker, can you evaluate no-CoT models via approximate proxies (e.g., self-consistency runs, early-exit triggering, or token-budgeted answers) and report at least Efficiency/Accuracy?

- Connection to “token complexity” trade-offs. The paper reports an efficiency–performance trade-off qualitatively (and via Efficiency vs. Tokens). It would be compelling to plot accuracy vs. total tokens (and vs. first-correct tokens) per model and per difficulty, and connect to the token complexity phenonmen in [1] to test curve universality on Think-Bench. The data seem available (Tables 2–6).

[1] Lee, A., Che, E., & Peng, T. (2025). How well do llms compress their own chain-of-thought? a token complexity approach. arXiv preprint arXiv:2503.01141.

---

> ### Author Response · Authors · 2025-11-21
> **Response to W1, W2 and Q1**
>
> **Response to W1 and Q1:**
>
> We sincerely appreciate you raising this important point concerning model coverage. We agree that the inclusion of closed-source models such as GPT-4/5 and Gemini-2.5-Pro would further enhance the external validity of our benchmark. However, these models currently do not provide access to their internal chain-of-thought or intermediate reasoning traces—even when utilizing the three proxy methods you mentioned. This prevents the application of our full evaluation protocol, particularly the Efficiency and Reflection Quality metrics that rely on visible thinking content.
>
> We have conducted supplementary experiments on these models and report Recall, Precision, and Accuracy based on their final outputs. A new subsection has been added in the Appendix to present these supplementary results and discuss the performance of such models under our benchmark. Additionally, we have included a token budget experiment to examine the impact of token budget on model outcomes.
>
> Details are as follows (lines 736-774):
>
> As shown in Table 7, we compare the Recall, Precision, and Accuracy of models for which the platform does not provide intermediate thinking content. Overall, Gemini 2.5 Pro achieves the best performance across all three metrics, demonstrating stable and reliable reasoning capability. In contrast, GPT-5 and o4-mini exhibit abnormally low Recall scores. Further analysis reveals that this phenomenon does not stem from insufficient reasoning ability. Instead, for some questions, these models return only the final answer without any reasoning steps that could be aligned with the annotated key steps. Consequently, their matched-step counts are systematically underestimated, leading to depressed Recall values.
>
> To explore the influence of output length, we conduct a token-budget study on GPT-5. Table 8 shows that increasing the token budget from 3000 to unlimited improves Accuracy from 75.85\% to 93.16\%. This demonstrates that limiting the token budget affects the completeness of the model’s outputs and consequently its measurable performance.
>
> |Model|Recall|Precision|Accuracy|
> |----|----|----|----|
> |Don't provide thinking content||||
> |GPT-5|70.65%|95.41%|93.16%|
> |o4-mini|73.25%|92.76%|95.42%|
> |Gemini2.5-pro|**88.46%**|**95.69%**|**97.16%**|
> |Provide thinking content||||
> |Claude-3.7-sonnet|81.29%|86.26%|93.89%|
> |Deepseek-r1-distill-qwen-1.5b|47.10%|59.61%|60.58%|
> |Deepseek-r1-distill-qwen-7b|63.65%|77.29%|81.53%|
> |Deepseek-r1-distill-qwen-14b|61.04%|79.97%|80.65%|
> |Deepseek-r1-distill-qwen-32b|64.17%|83.76%|83.71%|
> |Deepseek-r1|80.80%|88.33%|93.82%|
> |Ernie-x1-turbo-32k|82.03%|**88.67%**|92.36%|
> |Grok-3-mini-beta|81.56%|86.51%|93.96%|
> |Qwen3-235b-a22b|**85.80%**|86.97%|**94.98%**|
> |Qwq-plus|80.40%|85.08%|90.76%|
> |Glm-z1-air|80.16%|83.18%|91.92%|
>
> |Token Budget|Accuracy|
> |--------------|----------|
> |3000|75.85%|
> |5000|84.87%|
> |No limited|93.16%|
>
> **Response to W2:**
>
> Thank you for raising this concern. To verify the robustness and generality of the proposed metrics, we conducted an additional noise-injection experiment to test whether the evaluation pipeline—including the judge model and the proposed metrics—remains stable when Large Language Model (LLM) outputs are perturbed.
>
> We injected controlled noise into the reasoning traces of LLMs, including random word substitutions and deletions, to simulate paraphrase variations or noise interference scenarios. Subsequently, the noisy outputs were re-evaluated using the same judging process. The results demonstrate that the proposed metrics are not overly sensitive to surface-form heuristics and maintain robustness under realistic perturbation scenarios.
>
> Details are as follows (lines 1145-1154):
>
> As shown in Table 12, to assess the robustness of the judge model against real-world variations in LRM outputs, we conduct a noise disturbance experiment. We introduce controlled noise to LRM outputs through random word replacements and deletions, simulating common perturbations in generated content. We then evaluate the stability of judge model metrics under both original and noisy input conditions. The results demonstrate that Claude 3.7 Sonnet maintains remarkable stability when processing noisy LRM outputs. Its core evaluation metrics exhibit negligible fluctuations compared to the original input scenario. This resistance to minor output perturbations confirms that both Claude 3.7 Sonnet  and the designed evaluation metrics exhibit robust stability, reliably maintaining consistent evaluation standards even when faced with imperfect or slightly distorted LRM outputs.
>
> |Model|Thought Num-Original|Thought Num-Add noisy|Efficiency-Original|Efficiency-Add noisy|
> |-----|--------------------|---------------------|--------------------|---------------------|
> |Claude-3.7-Sonnet|0.28|0.27(-0.01)|49.61\%|50.12\%(+0.51\%)|
> |Deepseek-r1|9.17|9.17(0.0)|48.96\%|49.93\%(+0.93\%)|
> |Grok-3-mini-beta|0.38|0.37(-0.01)|61.69\%|62.64\%(+0.95\%)|

---

> ### Author Response · Authors · 2025-11-21
> **Response to W3 and W4**
>
> **Response to W3:**
>
> (a)Thank you for raising your concerns regarding the practicality and extensibility of our benchmark. First, although we adopted Claude 3.7 Sonnet as the judge model in our experiments, the evaluation pipeline proposed in this study is not dependent on any specific judge model. Any sufficiently powerful large language model can serve as an alternative judge, though potential fluctuations in experimental results may occur. Additionally, we have conducted relevant experiments on the selection of judge models, which are presented in the Appendix. As shown in the following table, Claude 3.7 Sonnet achieved results more consistent with human judgments, and thus we selected it as the judge model for our experiments.
>
> Detail in appendix E.2 and E.3 (lines 1118-1176).
> |Model|GPT-4o-mini|o3-mini|Qwen-2.5-72B-instruction|Claude3.7Sonnet|
> |-----|-----------|-------|------------------------|---------------|
> |IAA|0.9|0.93|0.88|0.95|
> |Kappa|0.79|0.81|0.74|0.85|
>
> (b)In terms of domain coverage.  To ensure the quality of human-verified key-step annotations, our study focuses on three STEM domains.  STEM problems are also widely viewed as the most direct way to evaluate reasoning ability, because they contain clear logical structure and unambiguous intermediate steps.  Recent reasoning benchmarks such as TTT Bench[1] similarly restrict their evaluation to STEM-style tasks.  Although extending to more subject areas is valuable, a single benchmark cannot cover all aspects of reasoning evaluation.  Our goal is to provide a high-quality, extensible foundation.  The benchmark’s annotation scheme and evaluation pipeline were designed to be modular, and future work or community contributions can easily extend it to additional domains.
>
> [1] Mishra, Prakamya, et al. "TTT-Bench: A Benchmark for Evaluating Reasoning Ability with Simple and Novel Tic-Tac-Toe-style Games." arXiv preprint arXiv:2506.10209 (2025).
>
> **Response to W4:**
>
> Thank you for highlighting the importance of quantifying annotation reliability and robustness. We conducted an additional evaluation on 300 randomly sampled items to assess the consistency of our human annotation reasoning steps. We report the resulting Cohen’s Kappa coefficien in Appendix E, which indicate substantial agreement across annotators. This provides empirical evidence that our step annotations are stable and reliable. The details are as follows (lines 1086-1115):
>
> **Annotation Consistency Evaluation**
>
> We recruit two graduate-level annotators to participate in the annotation consistency test. All annotators receive unified training on the annotation guidelines (detailed in Section 2.3), including the definition of key reasoning steps, annotation scope, and standard formatting requirements.
>
> **Experimental Design**
>
> A random sample of 300 instances is selected from Think-Bench. Each instance is annotated independently by A1 and A2. For each candidate step in an instance, both annotators label it as either "Key" or "Non-Key". After completing annotations for all 300 instances, we first count the number of instances where the two annotators agree on step labels, then use Cohen’s Kappa coefficient to quantify their consistency.
>
> **Results and Analysis**
>
> Table 8 summarizes the agreement and disagreement counts between A1 and A2 across the 300 instances. An "Agreed" instance means A1 and A2 share identical "Key/Non-Key" labels for all steps in the instance; a "Disagreed" instance means there is at least one step where their labels differ. Among the 300 instances, 252 are marked as Agreed, and 48 are marked as Disagreed. This results in an overall raw agreement rate of 94\% (282/300), indicating strong initial alignment between the two annotators, and the Cohen’s Kappa coefficient for their consistency is 0.878.
> |AnnotatorA|AnnotatorB-Key|AnnotatorB-Non-Key|
> |----------|--------------|------------------|
> |Key|120|10|
> |Non-Key|8|162|

---

> ### Author Response · Authors · 2025-11-21
> **Response to Q2**
>
> **Response to Q2:**
>
> We appreciate your insightful suggestion to more explicitly connect our findings to the token complexity phenomenon. Following the comment, we have added a dedicated analysis in the Appendix (see Figure 7, lines 810-910).
>
> In this extended analysis, we examine accuracy vs. total tokens as well as accuracy vs. first-correct tokens across models and difficulty levels. The results clearly exhibit a token complexity pattern consistent with prior observations~\citep{lee2025well}. Specifically, with the increase in problem difficulty and the model’s tendency toward in-depth verification, the total number of tokens exhibits a non-linear growth. Specifically, a large number of additional reflection or verification steps do not necessarily lead to a proportional improvement in accuracy. This trend is observed across multiple large language models, with accuracy plateauing in the high token consumption range. Second, when analyzing with First Correct Tokens as the horizontal axis, the curves of different models demonstrate the critical role of early correct convergence in reasoning efficiency. When the First Correct Tokens are low and the Efficiency is high, the model can significantly reduce the total computational cost while maintaining accuracy. In contrast, scenarios with both high First Correct Tokens and high Reflection Tokens often correspond to high accuracy but extremely high token complexity, a situation that is more prone to inducing overthinking especially for simple problems.

---

> ### Author Response · Authors · 2025-11-27
>
> Dear Reviewer oQGU,
>
> Thank you very much for the time and effort you dedicated to reviewing our work and for the valuable comments you provided. We carefully considered all your feedback, addressed the concerns in our response, and revised our paper accordingly.
>
> We sincerely hope that our clarifications and additional experiments help resolve your concerns. If you have any further concerns or questions, please let us know and we are more than happy to further address them.
>
> Thank you again for your thoughtful and constructive review.
>
> Best regards,
>
> The Authors

---

### Author Response · Authors · 2025-12-02
**Rebuttal Summary: Expanded Analyses on Model Coverage, Metric Robustness, Judge Reliability, Efficiency Definition & Error Analysis**

Dear Reviewers and AC,

We sincerely thank you for your time and constructive feedback. We have carefully addressed all reviewer concerns and updated the paper accordingly, with major clarifications and new experiments added in the revision, with changes highlighted in $\color{blue}{\text{blue}}$.

---

# **Summary**

We are encouraged that reviewers recognize the importance of studying reasoning efficiency and the need for *fine-grained, step-level evaluation*.
We have addressed key concerns regarding **model coverage**, **metric validity**, **judge reliability**, **multi-path reasoning**, **efficiency definition**, and **error analysis**, supported by extensive new experiments and clarifications.

## **Additional Experiments & Revisions**

- **Supplementary Evaluation of Closed-Source Models (oQGU)**
To address concerns on external validity, we added evaluations of GPT-5, Gemini-2.5-Pro, and o4-mini using output-based metrics (Accuracy, Recall, Precision), plus a token-budget study.
- **Noise Robustness of Metrics (oQGU)**
We injected controlled noise (random substitutions/deletions) into LRM reasoning traces to test metric and judge stability.
**Result**: All metrics remain stable with negligible fluctuations (Appendix Table 12), confirming robustness beyond surface-form heuristics.
- **Reliability of Judge Models (oQGU, gcSF, LYUk)**
We conducted comprehensive human–LLM agreement tests on 968 steps and compared multiple judges (GPT-4o-mini, o3-mini, Qwen-2.5-72B, Claude 3.7).
**Result**: Claude achieves the strongest alignment with human labels (95% agreement; κ = 0.85), validating its use as the primary judge.
- **Human Annotation Reliability & Multi-Path Coverage (oQGU, LYUk)**
We evaluated inter-annotator consistency on 300 samples and clarified that annotators labeled 2–4 representative solution paths per question.
**Result**: Strong human-human agreement (κ = 0.878) and sampling studies confirm that low scores arise from missing key steps—not unannotated but valid paths.
- **Comprehensive Token-Complexity & Trade-off Analysis (oQGU, fUgn)**
We added plots of accuracy vs. total tokens and accuracy vs. first-correct tokens, connecting them to token-complexity theory.
**Result**: The model shows a clear complexity curve and an efficiency-quality trade-off. The more difficult the problem, the more tokens are consumed.
- **Expanded Error-Type Analysis (fUgn, gcSF)**
We included broader categorization of reasoning failures (misinterpretation, boundary omission, flawed deductions).
**Result**: New examples illustrate systematic reasoning errors rather than model-side engineering failures.
- **Prompt-Based Overthinking Mitigation (oQGU, gcSF)**
We added a new experiment using minimal-thought prompting to reduce reasoning overhead.
**Result**: Token reductions up to 24% and measurable improvements in Efficiency validate Think-Bench’s ability to track mitigation strategies.

## **Key Clarifications**

- **Efficiency Definition (fUgn, gcSF)**
We clarified that Efficiency measures **intrinsic reasoning efficiency**, defined strictly as:
**First-Correct-Tokens / Total-Tokens,**
and is robust given real LRM behavior (“reason-then-answer” rather than “answer-then-verify”).
- **Treatment of Empty Outputs (gcSF)**
We clarified that **all empty outputs were counted as incorrect**, not skipped.
Failure rates are <0.5% and do not affect rankings.
- **Domain Scope & Extensibility (oQGU, gcSF)**
We explained that STEM domains were chosen for annotation quality, but the framework is modular and can be extended to other areas.
- **Cost & Reproducibility (fUgn, gcSF)**
We added a full cost report (≈ `$40` per model; total ≈ `$1,000`).
All data and scripts are released via the anonymous repository; open-source judges can eliminate evaluation cost.
- **Reflection Quality Criteria (gcSF, LYUk)**
We added explicit operational definitions and boundary examples distinguishing valid vs. invalid reflection.

---

We believe these revisions substantially strengthen the paper and demonstrate the robustness, fairness, and practicality of Think-Bench. **If there are any further questions or suggestions, we would be more than happy to address them.**

Best regards,

All Authors

---

### Meta-Review · Area_Chair_nkSH · 2026-01-03

**Summary:**

This work introduces Think-Bench, a benchmark designed to evaluate the reasoning efficiency of Large reasoning models (LRMs). Additionally, the authors propose some novel efficiency metrics and conduct a comprehensive evaluation of various LRMs across multiple dimensions, including the reasoning process, outcome quality, and chain-of-thought (CoT) characteristics. This work explores an interesting topic in LRMs evaluation domain. However, most reviewers raised concerns about the reliability and computational efficiency of the Claude 3.7 Sonnet–based evaluator used in this work, which affects the generalizability of this work. Overall, I think the paper falls below the acceptance threshold.

**Reviewer Scores:**

No reviewer updated their scores.

---

### Decision · Program_Chairs · 2026-01-26

Reject